# EXACT BYTE-LEVEL PROBABILITIES FROM TOKENIZED LANGUAGE MODELS FOR FIM-TASKS AND MODEL ENSEMBLES

**Buu Phan**[1*], **Brandon Amos**[2], **Itai Gat**[2], **Marton Havasi**[2], **Matthew Muckley**[2], **Karen Ullrich**[2]
[1]University of Toronto, [2]Meta AI
`truong.phan@mail.utoronto.ca`
`{bda, itaigat, marton, mmuckley, karenu}@meta.com`

## ABSTRACT

Tokenization is associated with many poorly understood shortcomings in language models (LMs), yet remains an important component for long sequence scaling purposes. This work studies how tokenization impacts model performance by analyzing and comparing the stochastic behavior of tokenized models with their byte-level, or token-free, counterparts. We discover that, even when the two models are statistically equivalent, their predictive distributions over the next byte can be substantially different, a phenomenon we term as "tokenization bias". To fully characterize this phenomenon, we introduce the Byte-Token Representation Lemma, a framework that establishes a mapping between the learned token distribution and its equivalent byte-level distribution. From this result, we develop a next-byte sampling algorithm that eliminates tokenization bias without requiring further training or optimization. In other words, this enables zero-shot conversion of tokenized LMs into statistically equivalent token-free ones. We demonstrate its broad applicability with two use cases: fill-in-the-middle (FIM) tasks and model ensembles. In FIM tasks where input prompts may terminate mid-token, leading to out-of-distribution tokenization, our method mitigates performance degradation and achieves 18% improvement in FIM coding benchmarks, while consistently outperforming the standard token healing fix. For model ensembles where each model employs a distinct vocabulary, our approach enables seamless integration, resulting in improved performance up to 3.7% over individual models across various standard baselines in reasoning, knowledge, and coding. Code is available at: https://github.com/facebookresearch/Exact-Byte-Level-Probabilities-from-Tokenized-LMs.

## 1 INTRODUCTION

Transformers form the backbone of all widely-used state-of-the-art language models (LMs) such as GPTs (Brown et al., 2020), Llama (Touvron et al., 2023), ans Mistral (Jiang et al., 2023a). A common pre-processing step in these models is tokenization, a method that shortens the input sequence by mapping multiple bytes, i.e. characters into discrete tokens, i.e. words or subwords, from a fixed vocabulary [1]. Efforts to bypass tokenization have shown limited empirical success (Yu et al., 2024; Limisiewicz et al., 2024), suggesting tokenization is critical to the performance of large language models (LLMs). It has been speculated that the reason for the performance gap between tokenized and byte-level models is due to the reduction of input tokens, allowing models to handle longer contexts at less compute (Zouhar et al., 2023; Goldman et al., 2024). Recent work by Rajaraman et al. (2024) provides an additional explanation: Even for unlimited data and compute [2], tokenized models can achieve better cross-entropy loss than untokenized ones, resulting in superior performance.

---

[*]Work done during internship at Meta AI.

[1]In the context of this study, we use the terms "character" and "byte" interchangeably to refer to an element from a subset of the tokenization vocabulary. This subset is somewhat flexible. Precision is only important in the experiment section where we define the subset to be all utf-8 bytes.

[2]And assuming the data source can be approximated as a $k^{th}$ order Markov chain.

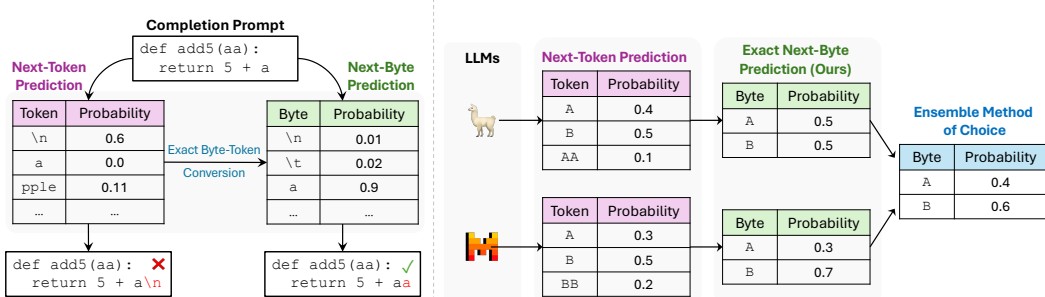

Figure 1: **Left:** Tokenized LMs can experience tokenization bias when prompts end mid-token, as in this code completion example. This means that the correct solution, a has zero probability of being chosen. Our method avoids this problem and can predict the correct token while using the same model. **Right:** Our method maps next-token predictions of arbitrary tokenized LMs to statistically equivalent next-byte predictions (see Section 4.1 for details). This enables any model ensemble strategy, such as averaging or mixture of experts.

To contribute to the understanding of the impact of tokenization, we demonstrate that for any tokenized LM, there exists a statistically equivalent byte-level process. Despite this fact, we find a surprising discrepancy in their predictive behaviors—particularly in next-token and next-byte predictions, where there can be significant differences. We hence introduce the concept of *tokenization bias* to describe the discrepancy between the predictive distributions of the tokenized model and the byte-level equivalent. Towards *mapping tokenized predictions to byte-level predictions*, we present the Byte-Token Representation Lemma (Section 4.1). This result enables both byte-level predictions and an algorithmic correction of tokenization bias in any trained LM without the need for additional training or optimization, allowing us to *sample bias-free next-bytes from any tokenized LM*.

Since tokenization bias limits certain token combinations and impairs the model's ability to sample specific tokens, it is especially detrimental for tasks such as coding, where precise completion is necessary. We focus on fill-in-the-middle (FIM) tasks, where tokenization bias poses significant challenges, particularly when a prompt ends mid-token. Previous research (Dagan et al., 2024) highlights this issue, showing that language models struggle to generate the correct completion in such cases. Our theory predicts this phenomenon precisely, and the failure example shown in Figure 1 (Left) can be reproduced with any open-weight model to date. Our experiments show that our next-byte prediction method outperforms tokenized models with the same cross-entropy by 18%, and is able to corrects cases where specialized fixes, such as token healing/alignment (Dagan et al., 2024; Roziere et al., 2023; Athiwaratkun et al., 2024), consistently fails to backup.

A byproduct of our byte-level prediction algorithm is that it enables ensembling of arbitrary LMs. Ordinarily, one cannot average the predictions of LMs that do not share a vocabulary. By mapping their predictions and conditioning domains to byte-space, one can easily aggregate predictions from multiple models and leverage the benefits of ensembling. Combining multiple models is advantageous because ensembles are generally more accurate and robust than any individual member (Hastie et al., 2009). We confirm this empirically: ensembling LMs with our byte-level predictions outperform individual models in many cases. Overall, the key contributions of this work are:

1. We convert tokenized LMs into statistically equivalent token-free models, and demonstrate that their predictive distributions differ. We define this discrepancy as tokenization bias.

2. We introduce a method that enables next-byte prediction for any tokenized language model to fully mitigate tokenization bias. This algorithm is applied at inference time and carries an O(1) computational cost in terms of model runs.

3. We present strong empirical evaluations on FIM benchmarks (18% improvement) and model ensemble tasks (up to 3.7% improvement), further demonstrating the effectiveness of our approach.

## 2 NOTATIONS AND SETUP

### 2.1 STRING AND BYTE-LEVEL LANGUAGE MODELS

We denote the alphabet set as $\mathcal{A}$ and its element character, or byte, as $x$. By default and consistent with real-world data, unless otherwise stated, we include the end of string <EOS> byte in $\mathcal{A}$. For any (finite) string $s$, we denote its substring from location $i$ to $j$ as $x_i^j := x_i x_{i+1}..x_j$. We view any byte-

level LM as a discrete stochastic process that defines the autoregressive probability $P(x_{n+1}|x_1^n)$ for all $n$. This defines an infinite (stochastic) sequence $\mathbf{x} = x_1 x_2...$ where $\mathbf{x} \in \mathcal{X}$. With the existence of <EOS>, we have $P(x_{n+1}|x_1^n) = 1.0$ for $x_{n+1} = x_n =$<EOS> and $P(x_{n+1} = $<EOS>$|x_1^n) \neq 0.0$ for any $x_1^n$. Effectively, $\mathbf{x}$ have finite length and the set $\mathcal{X}$ is countably infinite. We refer to $\mathbf{x}$ as a byte sequence instead of a string, which has a finite length and does not necessarily end with <EOS>.

Given $x_1^n$, the function $\text{prefix}(.)$ returns all possible prefixes of $x_1^n$, i.e. $\text{prefix}(x_1^n) = \{x_1^1, x_1^2, x_1^3, ..., x_1^n\}$. We denote the concatenation operation as $\text{concat}(.)$, i.e. $\text{concat}(x_1^{n_1}, y_1^{n_2}) = x_1...x_{n_1}y_1...y_{n_2}$. We define $\mathcal{X}(x_1^n)$ as the set of all byte sequences $\mathbf{x}$ with prefix $x_1^n$, i.e.

$$\mathcal{X}(x_1^n) = \{\mathbf{x}|x_1^n \in \text{prefix}(\mathbf{x})\},$$

which corresponds to the byte-level probability of obtaining a sequence with prefix $x_1^n$, or $P(x_1^n)$.

## 2.2 Tokenized Language Models

We consider two commonly used tokenization algorithms, namely Byte-Pair Encoding (BPE) (see Algorithm 2 in Appendix A.4) and Maximum Prefix Encoding (MPE). Tokens are elements within a vocabulary $\mathcal{V}$, where $\mathcal{A} \subseteq \mathcal{V}$, an individual token is denoted as $t \in \mathcal{V}$. We also assume <EOS> is not a part of any token but itself. An encoding consisting of $k$ tokens of a string $x_1^n$ is $t_1^k = \text{encode}(x_1^n)$. Conversely, a decoding of $t_1^k$ is a string denoted as $x_1^n = \text{decode}(t_1^k)$. Encodings with an unspecified number of tokens are denoted as $\vec{t}$ and $\vec{t}_{[i:j]}$ are the elements from $i$ to $j$ of $\vec{t}$. Note that BPE and MPE are deterministic, i.e. each string $s$ and sequence $\mathbf{x}$ correspond to a unique encoding. We define $\mathcal{X}(t_1^k)$ as a set consisting of all sequences whose encodings start with $t_1^k$, i.e.

$$\mathcal{X}(t_1^k) = \{\mathbf{x}|t_1^k = \text{encode}(\mathbf{x})_1^k\},$$

which corresponds to the token-level probability $P(t_1^k)$, i.e the probability of obtaining a sequence whose encoding starts with $t_1^k$. We also view any tokenized LM as an autoregressive process that defines $P(t_{k+1}|t_1^k)$ and refer to an infinite token sequence as $\mathbf{t} = t_1 t_2...$ and $\mathbf{t} \in \mathcal{T}$. Similar as the byte-level process, with the existence of <EOS>, $\mathbf{t}$ is effectively finite and $\mathcal{T}$ is countably infinite.

## 3 Language Models and Tokenization Bias

### 3.1 Statistical Equivalence Between Data Generating Processes

We begin by establishing the definition of statistical equivalence between two stochastic processes. Then, we show that byte-level LMs and their induced tokenized LMs are statistically equivalent.

**Definition 1.** (Statistical Equivalence) For a countably infinite set $\mathcal{X}$, the byte-level data generating processes $\mathcal{G}_1$ and $\mathcal{G}_2$ are statistically equivalent if and only if:

$$P_{\mathcal{G}_1}(\mathbf{x}) = P_{\mathcal{G}_2}(\mathbf{x}) \text{ for all } \mathbf{x} \in \mathcal{X},$$

i.e., the chance of sampling $\mathbf{x}$ are identical for processes $\mathcal{G}_1$ and $\mathcal{G}_2$, denoted by their subscripts.

We consider the following two stochastic processes:

- $\mathcal{G}_1$ : the ground truth byte-level language models $P(x_{n+1}|x_1^n)$.
- $\mathcal{G}_2$ : consists of the tokenized model $P(t_{k+1}|t_1^k)$ induced from the process $\mathcal{G}_1$ and tokenization. The process $\mathcal{G}_2$ generates sequence $\mathbf{x}$ by autoregressively sampling new $t_{k+1}$ and maps $t_{k+1}$ to its byte-level representation using the $\text{decode}(.)$ function.

Since every sequence $\mathbf{x}$ maps to a unique encoding $\mathbf{t}=\text{encode}(\mathbf{x})$ that again maps to the same $\mathbf{x}$, we have $P(\mathbf{x})=P(\mathbf{t})$, and thus $P_{\mathcal{G}_1}(\mathbf{x})=P_{\mathcal{G}_2}(\mathbf{x})$ or two processes are statistically equivalent. This correspondence is unique because $\mathbf{x}$ must end with <EOS>. Importantly, we note that in general, $P(x_1^n) \neq P(\text{encode}(x_1^n))$, as there exist other encodings in addition to $\text{encode}(x_1^n)$ sharing the same prefix $x_1^n$. Nevertheless, this establishment implies that tokenized LMs can be converted into a statistically equivalent byte-level counterpart.

### 3.2 Tokenization Bias

Despite their statistical equivalence, the generation behavior of tokenized and byte-level LMs can be significantly different when prompted with the same string. We characterize the tokenization bias phenomenon that describes this discrepancy between conditioning domains, i.e. bytes versus tokens.

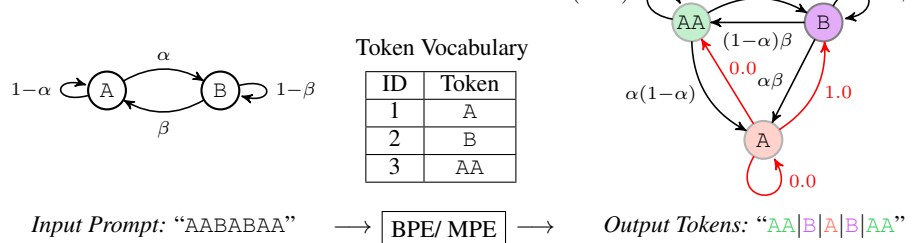

*Input Prompt:* "AABABAA" $\longrightarrow$ BPE/ MPE $\longrightarrow$ *Output Tokens:* "AA|B|A|B|AA"

Figure 2: Tokenization bias on a $1^{st}$-order Markov chain. Given the context token "A", the model will never sample the next token as "A", rather than with probability $1 - \alpha$. In practice, this bias occurs when prompts end with tokens that are part of another token, a common issue in the FIM tasks, leading to incorrect completions by the model.

**Definition 2.** (Tokenization Bias) Let the input prompt $x_1^n$ have the corresponding encoding $t_1^k = \text{encode}(x_1^n)$. The tokenization bias occurs for this prompt when:

$$P(x_{n+1}|x_1^n) \neq P(x_{n+1}|t_1^k), \tag{1}$$

where $P(x_{n+1}|t_1^k) = \sum_{t \in \mathcal{E}} P(t_{k+1} = t|t_1^k)$ and $\mathcal{E} = \{t \in \mathcal{V}|x_{n+1} = \text{decode}(t)_1\}$, i.e. the set of tokens whose first byte is $x_{n+1}$.

For example, the probability of the next byte being "c" may be different from the sum of the probabilities of all tokens that start with "c", which offers a broader perspective compared to the probability of the subsequent token being exactly "c".

### 3.2.1 TOKENIZATION BIAS IN MARKOV CHAINS

To systematically study tokenization bias, we employ a simplified autoregressive model, representing the data generating process as a Markov chain. As a result, we can derive a closed-form expression for the induced tokenized model and directly observe the tokenization bias phenomenon.

We consider a $1^{st}$ order Markov chain with two states {"A", "B"}, shown in Figure 2 (left) where each string is tokenized with either MPE or BPE, which yields the same result. With the vocabulary $\mathcal{V} = \{\text{"AA"}, \text{"A"}, \text{"B"}\}$, we obtain a new Markov chain whose states and transition matrix are shown in Figure 2 (right). Appendix A.7 provides details on computing the transition matrix. The statistical equivalency between two chains is described in Rajaraman et al. (2024). First, notice that no tokenization bias occurs when conditioning on $t_1 = \text{"AA"}$ or $t_1 = \text{"B"}$, e.g.

$$P(x_3 = \text{"A"}|x_1^2 = \text{"AA"}) = P(t_2 = \text{"AA"}|t_1 = \text{"AA"}) + P(t_2 = \text{"A"}|t_1 = \text{"AA"}) = 1 - \alpha$$

However, when conditioning on $t_1 = \text{"A"}$, tokenization bias emerges, i.e. the probability $P(x_2 = \text{"A"}|t_1 = \text{"A"}) = 0.0$ is not equal to $P(x_2 = \text{"A"}|x_1 = \text{"A"}) = 1 - \alpha$, i.e. the token-level Markov chain never samples "A" with any prompt ending with token "A". The reason is that whenever two consecutive bytes "A" appear together, the tokenizer immediately merges them into a single token "AA", results in zero probability. Despite being simple, this model portrays the exact phenomenon as the coding example discussed in Figure 1 (Left).

**Higher-Order Markov Chain.** We now show this phenomenon in a more complex, $3^{rd}$ order Markov chain, but shift from mathematical derivation to an empirical approach to demonstrate its relevance in practice. Furthermore, we leverage this example to illustrate the effectiveness of our byte-token conversion method, to be introduced in Section 4, as a viable bias correction technique. Here, we train a decoder transformer on tokenized data and analyse its next-token probabilities, shown in Figure 3. As we expect, the trained tokenized model exhibits severe bias in its predictions. Nevertheless, when combined with our bias correction technique, we

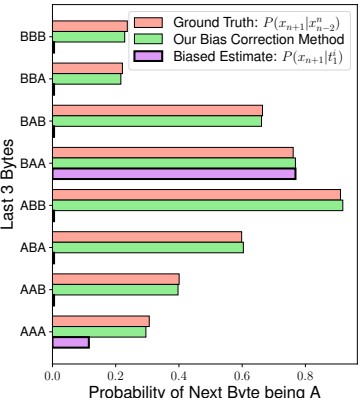

Figure 3: Tokenization bias on a $3^{rd}$ order Markov chain. Our byte-token conversion (bias correction) method in Section 4 accurately recovers $P(x_{n+1}|x_1^n)$ of the original chain.

accurately recover the original transition probabilities, demonstrating its potential for generalization to other LMs. We detail the experiment setup in Appendix A.7.

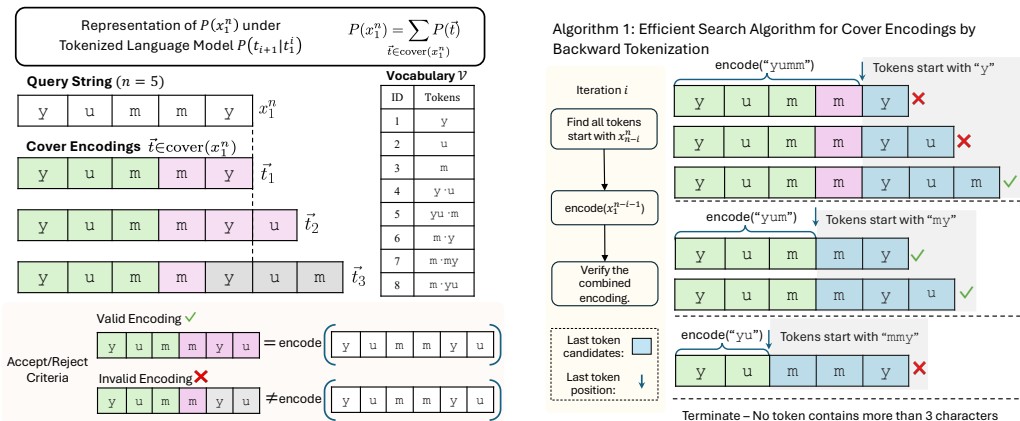

Figure 4: **Left:** Representation of $P(x_1^n)$ using tokenized LM with an example of cover encodings and valid/invalid encoding. **Right:** Illustration of Algorithm 1 for cover encodings search. Green tick and red cross denote valid and invalid encodings respectively (Definition 3). The termination step can be easily implemented despite not shown in the algorithm. We use BPE in this example.

### 3.2.2 INVALID ENCODINGS

The investigation on the zero probability events observed in our Markov chain example leads us to introduce the definition of invalid encodings, which sets constraints on the probability $P(\vec{t})$ of tokenized LMs. We remind the readers that this definition specifically targets BPE/MPE.

**Definition 3.** (Invalid Encodings) An encoding $\vec{t}$ is invalid under BPE/MPE encoding scheme if $\text{encode}(\text{decode}(\vec{t})) \neq \vec{t}$, else it is valid.

In other words, invalid encodings represent the string with different tokens than the one produced by the tokenizer. For example, assume MPE tokenizer and $\mathcal{V} = \{\text{"c"}, \text{"a"}, \text{"t"}, \text{"at"}, \text{"cat"}\}$, then $\text{encode}(\text{"catt"}) = [\text{"cat"}, \text{"t"}]$. Encodings $[\text{"c"}, \text{"at"}, \text{"t"}]$ and $[\text{"c"}, \text{"a"}, \text{"t"}, \text{"t"}]$ are invalid.

With this definition, Proposition 1 states that no invalid encodings will exist in the token distribution.

**Proposition 1.** $\mathcal{X}(t_1^k) = \varnothing$ if and only if $t_1^k$ is invalid. As a result, $P(t_1^k) = 0.0$. Furthermore, $P(t_k | t_1^{k-1}) = 0.0$ if $t_1^{k-1}$ is valid, otherwise, it is undefined.

*Proof. See Appendix A.4*

**Remark 1.** Proposition 1 indicates the constraints of $P(t_1^k)$ for tokenized LM induced by the byte-level data generating process. For trained LMs, $P(t_k | t_1^{k-1}) \neq 0$ for invalid $t_1^k$ due to softmax activations but we observe it to be very low compared to the valid $t_1^k$, similar to those in Figure 3. Since invalid encodings are provably non-existent in the ground-truth token-level distribution, we can truncate these invalid probabilities to zero without any loss in terms of perplexity score, detailed in Appendix A.6. From now, we assume that the LMs will assign zero probability to provably non-existent encodings in the token-level distribution.

## 4 EXACT BYTE-LEVEL PROBABILITIES

### 4.1 BYTE-TOKEN REPRESENTATION LEMMA FOR $P(x_1^n)$

To addresses the tokenization bias, we derive byte-level predictions $P(x_{n+1} | x_1^n)$ from token-level predictions $P(t_{k+1} | t_1^k)$. To this end, we compute $P(x_1^n)$ and begin with the concept of cover encodings of $x_1^n$. These encodings are valid encodings that "optimally" contain $x_1^n$, i.e. their last tokens start at locations $i < n$. Results in this section, i.e. Section 4.1, apply for any deterministic tokenizer, not just BPE/ MPE (also see Remark 1).

**Definition 4.** (Cover Encodings) Given a prefix $x_1^n$, an encoding $t_1^k$ is said to be covering $x_1^n$ when all the following conditions are satisfied:

1. $\mathcal{X}(t_1^k) \neq \varnothing$ (i.e. $t_1^k$ is a valid encoding in the case of BPE/MPE).

2. There exists an index $i$ such that $x_i^n \in \text{prefix}(\text{decode}(t_k))$ and $\text{decode}(t_1^{k-1}) = x_1^{i-1}$ where $1 \leq i \leq n$, i.e. $\text{decode}(t_1^k)$ starts with $x_1^n$ and the last token $t_k$ covers a suffix of $x_1^n$.

We denote $\mathrm{cover}(x_1^n)$ to be the set of all cover encodings of $x_1^n$, with examples in Figure 4 (mid-left).

We now establish the BTR Lemma that allows us to exactly compute $P(x_1^n)$ from $P(t_{k+1}|t_1^k)$. The main idea is that for any sequence $\mathbf{x}$ starting with $x_1^n$, its encoding $\mathrm{encode}(\mathbf{x})$ must start with one of the encodings in $\mathrm{cover}(x_1^n)$.

**Lemma 1.** *(Byte-Token Representation) For a language model $P(t_{k+1}|t_1^k)$, given a prefix $x_1^n$, the following statements hold:*

    *1. For any distinct $\vec{t}, \vec{t'} \in \mathrm{cover}(x_1^n)$, we have $\mathcal{X}(\vec{t}) \cap \mathcal{X}(\vec{t'}) = \varnothing$.*

    *2. $\mathcal{X}(x_1^n) = \bigcup\limits_{\vec{t} \in \mathrm{cover}(x_1^n)} \mathcal{X}(\vec{t})$.*

*As a result, $P(x_1^n)$ can be expressed as the marginal probability of all covering tokens of $x_1^n$*

$$P(x_1^n) = \sum_{\vec{t} \in \mathrm{cover}(x_1^n)} P(\vec{t}). \tag{2}$$

*Proof. See Appendix A.2.*

**Remark 2.** The BTR Lemma provides a general view of tokenization bias. Consider the cover encoding example in Figure 4 (middle-left). There, given the input string $x_1^5 = $ "yummy", prompting into the models $\mathrm{encode}(x_1^5) = [\text{"yum"}, \text{"my"}]$ will discards the other two possibilities, resulting in a skew distribution over the next-byte, i.e. tokenization bias.

Some scenarios requires conditioning on special tokens, such as when using control/synthetic tokens (e.g., FIM code-infilling). In such cases, we want to compute the byte-level probability conditioned on specific tokens, i.e. $P(x_{m+1}^n|t_1^k)$ where $x_1^m = \mathrm{decode}(t_1^k)$, or equivalently, $P(x_1^n, t_1^k)$. Corollary 1 provides a closed-form expression for this quantity.

**Corollary 1.** *We can express $P(x_1^n, t_1^k)$ where $\mathrm{decode}(t_1^k) \in \mathrm{prefix}(x_1^n)$ as follows:*

$$P(x_1^n, t_1^k) = \sum_{\substack{\vec{t'} \in \mathrm{cover}(x_1^n) \\ t_1^k = \vec{t'}_{[1:k]}}} P(\vec{t'}), \tag{3}$$

*i.e., to compute $P(x_1^n, t_1^k)$, we limit the set of cover encodings to those that starts with $t_1^k$.*

*Proof. See Appendix A.3.*

Finally, applying Lemma 1 requires searching all $\vec{t} \in \mathrm{cover}(x_1^n)$, which can be time-consuming with naive search. For the tokenizers under consideration, i.e. BPE and MPE, there exists an efficient algorithm by using the properties of invalid encodings (Definition 3), explained in Section 4.1.1.

### 4.1.1 COVER ENCODING SEARCH ALGORITHM

We present the cover search process for BPE/MPE in Algorithm 1, which is illustrated in Figure 4 (right). Note that the algorithm also returns $P(\vec{t})$ for $\vec{t} \in \mathrm{cover}(x_1^n)$ which we will use to sample subsequent characters later on. The idea is as follows: instead of searching for cover encodings from left to right, which can be computationally expensive, we search for valid encodings in reverse order, starting from the right. Suppose the last token $t_k$, of a cover encoding $t_1^k$ is known and it has suffix $x_{i+1}^n$, then according to Proposition 1, we must have: $t_1^{k-1} = \mathrm{encode}(x_1^i)$, else $t_1^k$ will be invalid.

As a result, searching from the reverse order, for any suffix $x_{i+1}^n$, we can find all tokens that start with $x_{i+1}^n$, tokenize $x_1^i$ and check if their combination is valid or not. The number of model runs is at most $n\ell$, where $\ell$ is length of the longest token in $\mathcal{V}$, in order to compute $P(\vec{t})$. In practice, the actual number of inference runs is much lower since $\mathrm{encode}(x_1^i)$ of the current iterations often contains encodings of the later iterations. Finally, while we can obtain $P(x_{n+1}|x_1^n)$ through factorization using this algorithm, it is not practical for sampling purpose as we need to repeat the process for all $x_{n+1} \in \mathcal{A}$. We next show an efficient alternative in Section 4.2.

---

**Algorithm 1** (Cover Encodings Search). Compute $P(x_1^n)$ and $P(\vec{t})$ for each $\vec{t} \in \text{cover}(x_1^n)$.

1: **procedure** EXTRACT_COVER($x_1^N$)
2:      cover_dict = {} *# Dictionary $\{\vec{t} : P(\vec{t})\}$*
3:      **for** $i = n - 1, ..., 0$ **do**
4:          *# Find all tokens covering $x_{i+1}^n$ and tokenize the remaining $x_1^i$*
5:          $\mathcal{B} = \{t \in \mathcal{V} | x_{i+1}^n \in \text{prefix}(\text{decode}(t))\}$
6:          $t_1^{k-1} = \text{encode}(x_1^i)$
7:          *# Compute $P(t_1^k)$ where $t_1^k = t_1, ... t_{k-1}, t$ for all $t \in \mathcal{V}$*
8:          $P(t_1^k) = P(t_1^{k-1}) \times P(t_{k+1} = t | t_1^{k-1})$ *# Broadcast Multiplication for all t*
9:          *# Remove invalid encodings.*
10:          **for** $t \in \mathcal{B}$ **do**
11:              $\vec{t} = \text{concat}(t_1^{k-1}, t)$
12:              cover_dict$[\vec{t}] = P(\vec{t})$ **if** is_valid($\vec{t}$)
13:          **end for**
14:      **end for**
15:      *# $\text{cover}(x_1^n) = \text{cover\_dict.keys}()$*
16:      $P(x_1^n) = \sum\limits_{\vec{t} \in \text{cover\_dict}} P(\vec{t})$ *# from the BTR Lemma ( Lemma 1).*
17:      **return** $P(x_1^n)$, cover_dict
18: **end procedure**

---

### 4.2   EFFICIENT NEXT-BYTE SAMPLING ALGORITHM FOR $P(x_{n+1} \mid x_1^n)$

To efficiently compute $P(x_{n+1} | x_1^n)$ for any $x_{n+1} = a \in \mathcal{A}$, we note that $\text{cover}(x_1^{n+1})$ contains:

- $\text{C}_{n+1}(a)$: encodings $\vec{t}$ from the previous $\text{cover}(x_1^n)$ whose $(n+1)^{\text{th}}$ bytes is $a$. Formally, we have: $\text{C}_{n+1}(a) = \{\vec{t} \in \text{cover}(x_1^n) | \text{decode}(\vec{t})_{n+1} = a\}$.

- $\bar{\text{C}}_{n+1}(a)$: encodings $\vec{t}$ whose last token starts with $a$ at the $(n+1)$ location. Formally, we have: $\bar{\text{C}}_{n+1}(a) = \{t_1^{k+1} | t_1^k = \text{encode}(x_1^n), \text{decode}(t_{k+1})_1 = a\}$.

Since $\text{C}_{n+1}(a) \cap \bar{\text{C}}_{n+1}(a) = \varnothing$, then:

$$P(x_1^n, x_{n+1} = a) = \sum_{\text{C}_{n+1}(a)} P(\vec{t}) + \sum_{\bar{\text{C}}_{n+1}(a)} P(\vec{t}), \tag{4}$$

With this formulation, we find $\text{cover}(x_1^{n+1})$ following the process in Figure 5 (also see Algorithm 3 in Appendix A.8). Specifically, to find $\text{C}_{n+1}(a)$ and $\bar{\text{C}}_{n+1}(a)$ for every $a \in \mathcal{A}$, we:

1. Obtain $\text{cover}(x_1^n)$ using Algorithm 1 or from Step 4 of the previous sample.

2. Find $\text{C}_{n+1}(a)$ by checking the $(n+1)^{th}$ byte of each $\vec{t} \in \text{cover}(x_1^n)$. We accumulate them for each $a$ respectively.

3. Find $\bar{\text{C}}_{n+1}(a)$ by querying the conditional distribution over all tokens $P(t_{k+1} | t_1^k)$. Note that $P(t_1^k)$ was already computed in $\text{cover}(x_1^n)$. The encoding accept/reject step is optional.

4. Obtain $\text{cover}(x_1^{n+1})$ for all $x_{n+1} = a \in \mathcal{A}$.

5. Sample $x_{n+1}$ from all computed $P(x_1^{n+1})$ (normalizing by $P(x_1^n)$).

Thus we only need to run Algorithm 1 at the beginning of the sampling process. In Step 3, the mapping of what tokens start with what bytes can be pre-computed and the sum over tokens can be parallelized via matrix multiplication. Also, one can avoid storing a large number of encodings in Step 4 by sampling the next byte immediately after Step 3, following Equation (4), and only create the set of the sampled value $x_{n+1}$, i.e. $\bar{\text{C}}_{n+1}(x_{n+1})$.

**Remark 3.** Computing $P(x_1^n)$ requires accessing to $P(t_1)$ but models such as the Yi series (Young et al., 2024) do not provide $P(t_1)$. Nevertheless, it is possible to compute $P(x_{n+1} | x_1^n)$ by leveraging the pre-tokenization pattern such as white spaces or punctuation. Details in Appendix A.1.

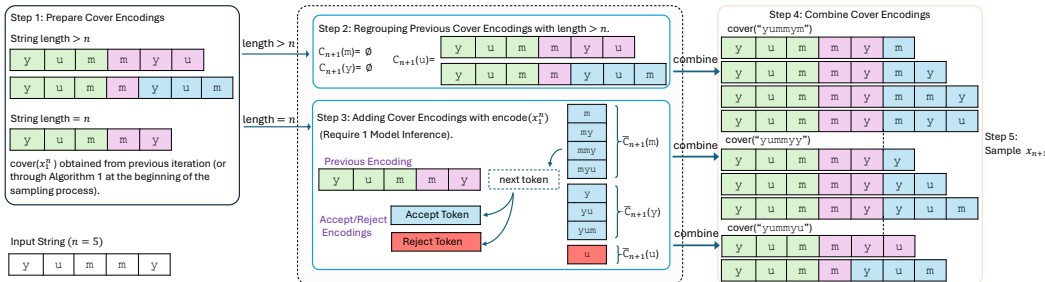

Figure 5: Illustration of the sampling process for $x_{n+1}$ from tokenized LM $P(t_{k+1}|t_1^k)$, with example following Figure 4. We do not include $P(\vec{t})$ for simplicity. Details in Section 4.2.

## 5 RELATED WORK

**Algorithms for Tokenization Bias.** Tokenization bias has been empirically observed when model produces unusual generations when prompted with string ends with trailing whitespace (Gao et al., 2023) or mid-word (Dagan et al., 2024). Token healing (Dagan et al., 2024; guidance ai, 2023) mitigates this issue by searching for tokens prefixed with the incomplete word. This method, however, fails to consider all matching possibilities. In the "yummy" example in Figure 4, it can only recover two encodings "yum|my" and "yum|myu" (tokens separated by "|") , as "my" is a prefix string of tokens "my" and "myu", missing "yum|m|yum" (also see Appendix D). Similar techniques such as token-alignment (Athiwaratkun et al., 2024) also failed to address all underlying issues (see Appendix E). Another proposed solution is using specific prompting strategies (Bavarian et al. (2022), also see PSM Mode in Section 6.1), requiring additional training and specific prompt structure. In contrast, we tackle the underlying issue by identifying the probabilistic root cause, i.e. the token-byte domain gap, and introduce a conversion technique to recover the unbiased byte-level distribution without requiring retraining. Finally, Pimentel & Meister (2024) address a different but related problem of computing next word probability, sharing some similar analysis but their approach is limited to whitespace-separated tokenizers and does not support autoregressive byte sampling.

**Language Model Ensembles** To the best of our knowledge, our work is the first to (i) compute next token probabilities in a universal space for both input and output domains while (ii) guaranteeing that the statistical properties of all member models are preserved (see Section 4), and (iii) without the need for additional training, data or model modifications. Several works also attempted to combine multiple LMs. Wan et al. (2024) introduce a model distillation technique by combining predictions from various LMs to fine-tune a primary model, which does not tackle the issue of vocabulary discrepancies. Jiang et al. (2023b) use ranking approach and evaluating outputs at the paragraph level and comparing them pairwise. However, their approach requires training a scoring function, making it sensitive to the distribution of training data and model choice. Huang et al. (2024) maps the token probabilities of member models to a universal space, which relies on token embedding's similarity and involves solving an optimization task. Gu et al. (2024) also investigate character-level ensembling, but does not provide expressions for $P(x_1^n)$ or $P(x_{n+1}|x_1^n)$ and overlooks the issue of invalid encodings. Since their method condition on tokens without addressing the bias, it is susceptible to invalid encoding states and division-by-zero errors, particularly with optimal sources.

## 6 EXPERIMENTS

### 6.1 CODE COMPLETION

Code completion with LLMs is well-suited for our byte-level predictions because the model is frequently prompted with incomplete code snippets. Here, we briefly introduce the setting and show that the byte-level predictions significantly improve upon the standard performance.

**Fill-in-the-middle (FIM) code completion.** Code completion is the process of taking source code as input along with a point in the middle to generate the completion, e.g., the point the user's cursor is at. So, generating new code in the middle does not fit into the standard autoregressive generation provided by pre-training models, nor does it fit into standard system-user-assistant conversation prompt templates. The solution has been to introduce a fill-in-the-middle (FIM) prompt template that turns the middle generation back into a standard prompt that can be completed by an autoregressive model (Bavarian et al., 2022). These FIM prompt templates introduce new tokens or control sequences and cannot be used with a model that is not aware of them, so code models are then trained (or fine-tuned) on a dataset of code formatted into these FIM prompt templates.

| Predictions | PSM | | SPM | |
|---|---|---|---|---|
| | Greedy | @10 | Greedy | @10 |
| Token-level | 64.9 | 84.0 | 45.0 | 66.5 |
| + healing | | | 62.7 | 83.8 |
| + alignment | | | 63.0 | 82.9 |
| Byte-level | **65.7** | **84.1** | 63.9 | **84.3** |

@10 here is the max over the temperature sweep on the right

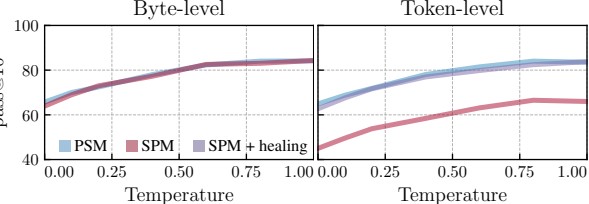

Figure 6: Pass rates for the HumanEval Random Span benchmark from Bavarian et al. (2022) with `CodeLlama-7b` for infilling Python. Across all settings, the byte-level predictions yield better completions. We evaluate the two standard infilling prompt templates of `Prefix-Suffix-Middle` (PSM) and `Suffix-Prefix-Middle` (SPM), and as baselines use token healing (Dagan et al., 2024) and alignment (Athiwaratkun et al., 2024). The SPM template forces the model to make predictions from the middle of a token, which our byte-level predictions successfully handle.

**Prompt templates for FIM.** Given a `{prefix}` and `{suffix}` along with the control tokens `<PRE>`, `<MID>`, and `<SUF>`, there are two main prompt formats for querying a code model to generate the middle portion: the `Prefix-Suffix-Middle` (PSM) format `"<PRE> {prefix} <SUF>{suffix} <MID>"`, and `Suffix-Prefix-Middle` (SPM) format is `"<PRE> <SUF>{suffix} <MID> {prefix}"`.

**Upsides and downsides of PSM and SPM.** The default and most widely-used mode is PSM (Bavarian et al., 2022; Roziere et al., 2023). We speculate this is the default mode because there are no tokenization issues in it: the incomplete prefix is clearly separated from the middle portion to generate by the control tokens. While the PSM format solves tokenization issues with this separation, the downside is that it creates an unnatural separation between the prefix and middle generation. SPM mode overcomes this by prompting the model to generate immediately after the prefix, without any separating tokens. This contiguity is desirable as it would leverage more knowledge from the pretraining task, but now creates a tokenization issue when the prefix stops in the middle of a token. A standard heuristic to overcoming this in SPM mode is called *token healing*, e.g., as described in Dagan et al. (2024), but it does not correct every tokenization issue here. Another fix is token-alignment (Athiwaratkun et al., 2024) but it is biased and suboptimal (see Appendix E). These issues make the PSM template outperform the SPM template in every previous results for code infilling.

**Our results.** We experimented with the standard random-span infilling benchmark from (Bavarian et al., 2022). Figure 6 shows that our byte-level predictions in SPM mode attain superior performance when using 10 samples, using `CodeLlama-7b`. This corroborates our hypothesis that the contiguity between the prompted prefix and generated middle in SPM mode is more natural and consistent with the pre-training objective of the model. Another surprising result is that the byte-level greedy generations outperform the token-level greedy generations.

**Ablation** This is surprising because each token typically spans over multiple bytes, hence for greedy generation the token model should have a slight advantage. We wanted to further investigate this finding and hence performed beam search with beam width 2 and 4, which led to consistent improvement of up to 2.9% and best performance at 67.7. The complete ablation is located in Appendix C.

## 6.2 MODEL ENSEMBLES

**Ensemble methods** rely on two principles; model prediction averaging (aggregation) and training each member of a model ensemble on a subset of all available data (bootstrapping). Ensemble methods are a popular choice because aggregation reduces the variance of the expected empirical test error, and bootstrapping reduces model bias. As a result, bootstrap aggregation leads to model ensembles that are generally more accurate and robust than any individual member model (Hastie et al., 2009). The recent surge of open-weights LMs, each presumably trained on different subsets of available text, makes these LLMs ideal candidates for bootstrap aggregation.

**Our method.** Unfortunately, vocabulary discrepancies between LMs prevent direct aggregation, as models map to different token spaces. Our method solves this by enabling exact next-byte prediction, allowing any LM to map into the same space, so byte probabilities can be aggregated without restrictions on the aggregation function and preserving the statistical properties of the member models. In this work, we choose simple averaging. We leave more complex methods for future work.

| Model | MMLU Token | MMLU Byte | PIQA Token | PIQA Byte | NQ Token | NQ Byte | TriviaQA Token | TriviaQA Byte | GSM8K Token | GSM8K Byte |
|---|---|---|---|---|---|---|---|---|---|---|
| `LLama2-7B` | 45.4 | 45.7 | 78.1 | 78.2 | 25.0 | 23.4 | 58.4 | 58.1 | 15.1 | 12.5 |
| `Yi-1.5-6B` | 63.4 | 63.4 | 78.5 | 78.5 | 22.8 | 22.7 | 53.7 | 53.4 | 61.3 | **61.5** |
| `Mistral-7B-v0.3` | 62.1 | 62.1 | 80.1 | 80.3 | 28.5 | 28.8 | 63.6 | 63.5 | 39.0 | 39.0 |
| Voting (top-2) | | 62.1 | | 80.1 | | | | | | |
| Top-2 ensemble (Our) | | **65.4** | | **80.7** | | **30.0** | | **64.2** | | 55.8 |

Table 1: We evaluate the token and equivalent byte-level model performance of various open source LMs. We further show the byte-ensemble performance of the top-2 performing models. For all benchmarks but GSM8K, byte-level ensembles outperform single models and voting.

**Benchmarks and setup.** We compare next-token, next-byte, voting and byte-ensembles against one another on multiple knowledge, reasoning, and coding benchmarks. We chose our benchmarks due to their prevalence in testing general LM ability. Due to computational constraints, we test on 7B models primarily. Note that multiple choice tasks require likelihood evaluation (MMLU, PIQA) while other tasks require mid to long generations. Finally, voting is not trivially extended to reasoning tasks without training an evaluator network.

| Model | Human Eval @1 Token | Human Eval @1 Byte | MBPP@1 Token | MBPP@1 Byte |
|---|---|---|---|---|
| `CodeLlama2-7b` | 32.3 | 28.7 | 40.6 | 42.4 |
| `Codellama2-13b` | 35.8 | 33.5 | 47.6 | 47.4 |
| `Yi-Coder-1.5B` | 38.4 | 36.5 | 52.8 | **53.6** |
| Top-2 Ensemble | | **42.1** | | **53.6** |

Table 2: Token and byte-level performance of open-weight code LMs. The byte-ensemble performance of the top-2 models achieves the best results on the Human Eval benchmark while matching Yi-Coder-1.5B on MBPP.

**Our results.** Results are shown in Table 1 and 2. Byte and token predictions are on par with one another. Ensembling the top-2 performing models consistently lead to performance boost except when member model performance is too divergent, as in the case of GSM8K dataset where `Yi-1.5-6B` outperforms `Mistral-7B-v0.3`, the second best LM, by a margin of 20%. For coding tasks, the ensemble of `Yi-Coder-1.5B` and `CodeLlama2-13b` improves up to 3.7% on Human Eval dataset. Overall, these results showcase ensemble as an exciting direction for LMs collaboration.

# 7 CONCLUSION

**Conclusion and limitations.** This work shows that tokenized LMs have different predictive distributions than their statistically equivalent byte-level counterpart, i.e. tokenization bias. We introduced an O(1) next-byte prediction algorithm to mitigate this bias at inference time and showed its empirical relevance for FIM benchmarks and model ensemble tasks. Yet, our method introduces notable memory consumption in its current implementation, and incurs additional linear computational costs (one token = multiple bytes). At present, these factors may limit its practical application in resource-constrained environments. While our work provides new insights into the effects of tokenization, it can not explain or mitigate all related phenomena such as poor performance on arithmetic tasks.

**Future work.** While this work sheds some light on token- and byte-level LMs, several open questions remain. Notably, our theory does not provide insight into the employed greedy evaluation process, nor do we examine the cumulative impact of tokenization bias. Regarding the latter, our analysis is limited to single-token predictions, while real-world applications require generating hundreds of tokens. Building on our findings, future research can explore broader topics, such as bias-variance decomposition in LM ensembles, or interplays between bootstrapping and scaling laws. Another direction is model distillation, where larger models with larger vocabularies could be distilled into smaller models with specialized tokenization. Our work can also enable prompt optimization, such as for universal adversarial attacks, and facilitates relative uncertainty estimation. Also, this work only scratched the surface of the possibility of model ensembles. For example, models trained separately on different languages may outperform a single model trained on all languages simultaneously, as they can achieve more optimal entropy bounds by leveraging language-specific tokenizations. This might also improve the representation of otherwise underrepresented languages in a training data corpus. Finally, our results also open up new possibilities for tackling mechanism design problems (Duetting et al., 2024) involving different language models, where the tokenization bias issue can potentially create unfair advantages for certain models.

# 8 ACKNOWLEDGMENTS

We thank Shubham Toshniwal and Jack Lanchantin for early comments and discussions.

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

## A APPENDIX

### A.1 ESTIMATE $P(x_{n+1}|x_1^n)$ WHEN $P(t_1)$ IS NOT AVAILABLE

In practice, several models such as the Yi series do not provide $P(t_1)$ since it does not include the beginning of sequence token. As a result, we cannot compute $P(\vec{t})$. However, we can still compute $P(x_{n+1}|x_1^n)$ since their tokenization algorithm often force splitting words by using pre-tokenization pattern such as the whitespace. Let $x_i$ be the last whitespace byte in $x_1^n$ such that $x_{i-1}$ is not a whitespace. Then we know that $t_1^k = \text{encode}(x_1^i)$ must be the prefix encoding of all cover encodings of $x_1^n$. This holds not only for $x_1^n$ but also any string $s = \text{concat}(x_1^n, s')$ where $s'$ is an arbitrary suffix.

As such, using the BTR Lemma 1, we have:

$$P(x_1^n) = \sum_{\vec{t} \in \text{cover}(x_1^n)} P(\vec{t}) = P(t_1^k) \times \sum_{\vec{t'} \in \text{cover}(x_{i+1}^n)} P(\vec{t'}|t_1^k) \tag{5}$$

and hence, with factorization:

$$P(x_{n+1}|x_1^n) = \left( \sum_{\vec{t'} \in \text{cover}(x_{i+1}^{n+1})} P(\vec{t'}|t_1^k) \right) \Big/ \left( \sum_{\vec{t'} \in \text{cover}(x_{i+1}^n)} P(\vec{t'}|t_1^k) \right), \tag{6}$$

which shows that we do not need $P(t_1^k)$ to compute $P(x_1^n)$, since conditioning on them is sufficient. For autoregressive byte generation, we condition each probability term in Algorithm 3 with $t_1^k$.

### A.2 PROOF OF LEMMA 1

**Lemma 1.** (Byte-Token Representation Lemma) For a consistent tokenizer and a corresponding language model $P(t_{k+1}|t_1^k)$, given a prefix $x_1^n$, we have the followings:

1. For any distinct $\vec{t}, \vec{t'} \in \text{cover}(x_1^n)$, then $\mathcal{X}(\vec{t}) \cap \mathcal{X}(\vec{t'}) = \varnothing$.
2. $\mathcal{X}(x_1^n) = \bigcup_{\vec{t} \in \text{cover}(x_1^n)} \mathcal{X}(\vec{t})$.

As a result, $P(x_1^n)$ can be expressed as the marginal probability of all covering tokens of $x_1^n$

$$P(x_1^n) = \sum_{\vec{t} \in \text{cover}(x_1^n)} P(\vec{t}). \tag{7}$$

*Proof.* We prove each point as follows:

1. Proof by contradiction, let $t_1^i, t_1'^j \in \text{cover}(x_1^n)$ and $t_1^i \neq t_1'^j$. Suppose that there exists a sequence **x** where $t_1^i = \text{encode}(\mathbf{x})_1^i$ and $t_1'^j = \text{encode}(\mathbf{x})_1^j$. Without the loss of generalization, suppose $i < j$, then $t_1^i = t_1'^i$ since our tokenizer is deterministic. Hence, $t_1'^j$ cannot be a cover encoding of $x_1^n$.

2. This follows the definition of cover encodings.

Since each $\mathcal{X}(\vec{t})$ is pair-wise disjoint, we arrive at the final equation. $\square$

### A.3 PROOF OF COROLLARY 1

**Corollary 1.** We can express $P(x_1^n, t_1^k)$ where $\text{decode}(t_1^k) \in \text{prefix}(x_1^n)$ as follows:

$$P(x_1^n, t_1^k) = \sum_{\substack{\vec{t'} \in \text{cover}(x_1^n) \\ t_1^k = \vec{t'}_{[1:k]}}} P(\vec{t'}). \tag{8}$$

---

**Algorithm 2** Byte Pair Encoding Algorithm.

---

1: **procedure** ENCODE_BPE($x_1^N$, $\mathcal{V}$)
2:     *# Set initial encodings:*
3:     c_tokens $= x_1^N$
4:     *# Iterate over merging order in $\mathcal{V}$, the first $|\mathcal{A}|$ entries correspond the the alphabet (no merge happens):*
5:     **for** $i = |\mathcal{A}| + 1, ... |\mathcal{V}|$ **do**
6:         c_tokens $\leftarrow$ find_merge(c_tokens, $\mathcal{V}[i]$)
7:     **end for**
8:     **return** c_tokens
9: **end procedure**
10:
11: **procedure** find_merge(c_tokens, $v$)
12:     *# Left and right tokens for merging*
13:     $t_{\text{left}}, t_{\text{right}}, t_{\text{new}} = v[1], v[2], v[3]$
14:     old_tokens $=$ c_tokens
15:     new_tokens $= []$
16:     *# Find and merge tokens from left to right*
17:     $j = 1$
18:     **while** $j < |\text{old\_tokens}|$ **do**
19:         **if** old_tokens$[i, i+1] = t_{\text{left}}, t_{\text{right}}$ **then**
20:             new_tokens.append($t_{\text{new}}$)
21:             $j = j + 2$
22:         **else**
23:             new_tokens.append(old_tokens$[i]$)
24:             $j = j + 1$
25:         **end if**
26:     **end while**
27:     **return** new_tokens
28: **end procedure**

---

*Proof.* The proof follows the one for the BTR Lemma 1, where we only consider the cover encodings that start with $t_1^k$. $\qquad\square$

### A.4   PROOF OF PROPOSITION 1

For completeness, we first show the BPE encoding algorithm in Algorithm 2. For the MPE algorithm, the rule is greedily looking for the longest token that matches the prefix of the given text.

**Proposition 1.** $\mathcal{X}(t_1^k) = \varnothing$ if and only if $t_1^k$ is invalid.

*Proof.* For the case of BPE, we prove each direction as follows.

- If $\mathcal{X}(t_1^k) = \varnothing$ then $t_1^k$ is invalid: Since $\mathcal{X}(t_1^k) = \varnothing$, we know that there exist no sequence $\mathbf{x}$ such that $\text{encode}(\mathbf{x})_1^k = t_1^k$. This means there is also no string $s$ that satisfy $\text{encode}(s)_1^k = t_1^k$. As such, for $s = \text{decode}(t_1^k)$, we do not have $\text{encode}(\text{decode}(t_1^k)) = t_1^k$, which proves the result.

- If $t_1^k$ is invalid then $\mathcal{X}(t_1^k) = \varnothing$: Let $x_1^n = \text{decode}(t_1^k)$, it is sufficient to consider two string $s_1$ and $s_2$ that both have prefix $x_1^n$. Furthermore, we assume the first $i$ tokens of $s_1$ covers exactly $x_1^n$, i.e. $x_1^n = \text{decode}(t_1^i)$ and similarly, the first $j$ tokens of $s_2$ covers exactly $x_1^n$, i.e. $x_1^n = \text{decode}(t_1'^j)$. Then:

    1. Proving that invalid $t_1^k$ leads to $\mathcal{X}(t_1^k) = \varnothing$ is equivalently to proving $t_1^i = t_1'^j$ for any $s_1, s_2$.
    2. Re-running the BPE algorithm for $s_1$ and $s_2$ in parallel, we know that there will be no merge between any suffix of $x_1^n$ and the rest of strings, i.e. $s_1 \backslash x_1^n$ and $s_2 \backslash x_1^n$ due to the condition above (See Algorithm 2, line 6).
    3. Furthermore, for $s_1$, any time a merge happens within $x_1^n$ then the same merge must also happen within $x_1^n$ for $s_2$ and vice versa.

As the result, we have $t_1^i = t_1'^j$ and they must be equal to $\mathrm{encode}(x_1^n)$.

For the case of MPE, the proof is similar:

- If $\mathcal{X}(t_1^k) = \varnothing$ then $t_1^k$ is invalid: Since $\mathcal{X}(t_1^k) = \varnothing$, similar to the case of BPE, we know that there exist no string $s$ such that $\mathrm{encode}(s)_1^k = t_1^k$. As such, for $s = \mathrm{decode}(t_1^k)$, we do not have $\mathrm{encode}(\mathrm{decode}(t_1^k)) = t_1^k$, which proves the result.

- If $t_1^k$ is invalid then $\mathcal{X}(t_1^k) = \varnothing$: Let $x_1^n = \mathrm{decode}(t_1^k)$, we consider two string $s_1$ and $s_2$ that both have prefix $x_1^n$. Furthermore, we assume the first $i$ tokens of $s_1$ covers exactly $x_1^n$, i.e. $x_1^n = \mathrm{decode}(t_1^i)$ and similarly, the first $j$ tokens of $s_2$ covers exactly $x_1^n$, i.e. $x_1^n = \mathrm{decode}(t_1'^j)$. Since the MPE tokenizer will greedily looking for the longest token, hence one of the the encodings must not follow MPE encoding rule (contradiction).

This concludes the proof. □

## A.5 TOKENIZATION BIAS UNDER STOCHASTIC TOKENIZERS

Another class of tokenizers is non-deterministic, such as BPE Dropout (Provilkov et al., 2020), to address model's weakness against text fragmentation by randomly omitting tokens before text processing, theoretically training the model in multiple vocabularies. Although intended to enhance robustness, our analysis suggests that tokenization bias still exists despite less obvious. It is partially mitigated, nevertheless, since the model needs to minimize loss across varied vocabulary sets. The BTR lemma (Lemma 1) holds, requiring consideration of all valid encodings under this variability. In practice, given the substantial size of the vocabulary, BPE-dropout is expected to be robust to tokenization bias, i.e. assigning non-zero probabilities to invalid encodings. Yet, it is not commonly employed in training large-scale LLMs because the randomized tokenization introduces a computational bottleneck, which slows down the training process. Also, it may require greater model capacity to handle the increased complexity.

We demonstrate the tokenization bias in BPE-droupt in the following experiment. Here, we use the 1st order Markov chain example as in Figure 2 where $(\alpha, \beta) = (0.4, 0.3)$. We train a LLM with BPE dropout where a sequence can be tokenized either with the vocabulary $\mathcal{V}_1 = \{$"A", "B", "AA"$\}$ or $\mathcal{V}_2 = \{$"A", "B"$\}$ with equal probability. Consider the following input tokens (separated by |) and their next-token probabilities:

- "|B|A|" and next token probabilities $\{$"A" : 0.43, "B" : 0.57, "AA" : $10^{-5}\}$: tokenization bias happens according to Definition 2 but the probability does not go to 0.0 as the cross entropy loss is distributed between the two vocabularies.

- "|B|A|A|" and next token probabilities $\{$"A" : 0.59, "B" : 0.41, "AA" : $10^{-5}\}$: which approximately equal to the original byte-level probabilities since the LLM detects that the only possible vocabulary that produces these tokens is $\mathcal{V}_2 = \{$"A", "B"$\}$.

- "|B|AA|A|" and next token probabilities $\{$"A" : $10^{-4}$, "B" : 0.99, "AA" : $10^{-5}\}$: tokenization bias occurs, i.e. the LM only outputs token "B". Since "AA" only occurs in the vocabulary $\mathcal{V}_1 = \{$"A", "B", "AA"$\}$, the LLM detects the tokenization pattern and infer that the sequence must be tokenized with this vocabulary.

Finally, for the first case "BA", we can recover the exact probability by using the BTR lemma. In this case, its cover encoding and associated probabilities are $t_1 = $|B|A|$, t_2 = $|B|AA| and $P(t_1) = 0.104$ and $P(t_2) = 0.0415$. For the string "BAA", the first cover encoding is $t_2$, which is already computed and the other one is $t_3 = $|B|A|A| and $P(t_3 = 0.0443)$. Using factorization, we obtain $\alpha = 0.405$, which is approximately equal the the original value of $0.4$.

## A.6 ON PREDICTIVE DISTRIBUTION OF LANGUAGE MODELS

In practice, LMs trained with BPE/MPE often do not strictly follow Proposition 1 due to softmax activations. In this section, we show that given any tokenized LM, we can force its output probabilities to obey Proposition 1, without any loss in terms of perplexity score on the token domain. In

other words, we can turn a tokenized language model that does not follow Proposition 1 to the one that does while guaranteeing that the new model will always result in a lower token-level perplexity score.

We first introduce Proposition 2. In this proposition, we are given a target discrete probability distribution $p$ where we know some of the values will not happen, says $\Phi^*$. Assume that we have another distribution $q$ that approximates $p$, then we can produce another distribution $q^*$ that is closer to $p$ in terms of KL divergence by setting corresponding probabilities of $q$ in $\Phi^*$ to 0.0 and renormalize it.

**Proposition 2.** *Given a discrete distribution $p = \{p_1, p_2, ..., p_m\}$ and $q = \{q_1, q_2, ..., q_m\}$ with $q_i > 0.0$ for all $i$. Let $\Phi = \{i \in \mathbb{Z} | p_i = 0.0\}$ and $\Phi^* \subseteq \Phi$, we define $q^* = \{q_1^*, q_2^*, ..., q_m^*\}$ where $q_i^* = 0.0$ for $i \in \Phi^*$, and $q_j^* = q_j/(\sum_{l \notin \Phi^*} q_l)$. Then we have:*

$$D_{\mathrm{KL}}(p||q^*) \leq D_{\mathrm{KL}}(p||q), \tag{9}$$

*which implies that $q^*$ is closer to $p$ than $q$. We refer to the process of producing $q^*$ as truncate-renormalization (TR).*

*Proof.* Let $Z = (\sum_{l \notin \Phi} q_l)$ is the normalizing factor in $q^*$. Note that $Z \leq 1$ and as such $\log(Z) \leq 0$. Then:

$$D_{\mathrm{KL}}(p||q^*) = \sum_i p_i \log\left(\frac{p_i}{q_i^*}\right) \tag{10}$$

$$= \sum_{i \notin \Phi^*} p_i \log\left(\frac{p_i}{q_i^*}\right) \quad \text{, use } 0\log 0 = 0.0 \tag{11}$$

$$= \sum_{i \notin \Phi^*} p_i \log\left(\frac{p_i}{q_i/Z}\right) \tag{12}$$

$$= \left[\sum_{i \notin \Phi^*} p_i \log\left(\frac{p_i}{q_i}\right)\right] + \log(Z) \tag{13}$$

$$\leq \sum_{i \notin \Phi^*} p_i \log\left(\frac{p_i}{q_i}\right) = D_{\mathrm{KL}}(p||q), \tag{14}$$

which completes the proof. $\square$

Applying to our scenario, for any autoregressive LM $P_1(t_{k+1}|t_1^k)$ that does not follow Proposition 1 (due to the softmax activations), we can perform the TR process (since we know which encoding is invalid) to obtain a new LM $P_2(t_{k+1}|t_1^k)$, which is guaranteed to better approximate the ground-truth model $P(t_{k+1}|t_1^k)$. Thus, we are guaranteed that the token-level perplexity score of $P_2(t_{k+1}|t_1^k)$ is always lower than or equal to $P_1(t_{k+1}|t_1^k)$. Finally, in practice, we observe that the conditional $P_1(t_{k+1}|t_1^k)$ for invalid $t_1^{k+1}$ is significantly smaller than the valid tokens.

## A.7 THE MARKOV CHAIN EXAMPLE

We provide a detail computation of the Markov chain example in the main paper. Recall that in the original chain (in the character domain), we have the following:

$$P(x_2 = \text{``A''}|x_1 = \text{``A''}) = 1 - \alpha \tag{15}$$
$$P(x_2 = \text{``B''}|x_1 = \text{``A''}) = \alpha \tag{16}$$
$$P(x_2 = \text{``A''}|x_1 = \text{``B''}) = \beta \tag{17}$$
$$P(x_2 = \text{``B''}|x_1 = \text{``B''}) = 1 - \beta \tag{18}$$

We also assume the initial probability $\pi = \{\gamma, 1 - \gamma\}$ for "A" and "B" respectively. In the token domain, let first compute $P(t_2 = \text{``A''}|t_1 = \text{``AA''})$ where we have:

$$P(t_2 = \text{``A''}|t_1 = \text{``AA''}) = P(x_3^6 = \text{``ABA''}|x_1^2 = \text{``AA''}) + P(x_3^6 = \text{``ABB''}|x_1^2 = \text{``AA''}) \tag{19}$$
$$= P(x_3^5 = \text{``AB''}|x_1^2 = \text{``AA''}) \tag{20}$$
$$= \alpha(1 - \alpha), \tag{21}$$

where in the first equality, we do not include the case $x_3^6 = $ "AAA" and $x_3^6 = $ "AAB" since encode("AAA")$_1 = $ "AA" and encode("AAB")$_1 = $ "AA", which are not the token "A" that we are interested in. For other tokens when $t_1 = $ "B", the computation follows the same arguments.

Finally, for the case of $t_1 = $ "A", we note that it implies the second token must be "B" since any "A" after the first "A" must be tokenized into "AA" (invalid tokens). Hence, we have $P(t_2 = $ "B"$|t_1 = $ "A"$) = 1.0$. Finally, in this specific Markov chain, since order of the Markov chain in the character domain is 1, we do not need to consider the higher order of the Markov chain in the token domain.

**Higher-Order Markov Chain Experiment Setup.** We validate the estimated byte-level probabilities of the BTR Lemma (Lemma 1) using the Markov chain's transition probabilities as ground truth. Specifically, we simulate the data generating process as a 3rd order Markov chain with two states $\mathcal{A} = \{$"A","B"$\}$, where we randomly construct the transition matrix and the vocabulary $\mathcal{V} = \{$"A", "B", "B·A", "BA·A", "B·BAA", "A·A", "BA·BA", "B·B"$\}$. Here, the order within $\mathcal{V}$ is the merging order for the BPE encoding process and the "·" separates the merging tokens. We then train a LM model using GPT-2 architecture with 6 hidden layers on the synthetically generated data. Since the model is agnostic to the Markov chain order, we average the probability from 100 runs on different context length while fixing the last 3 bytes.

### A.8    Next-Byte Sampling Algorithm

Algorithm 3 shows the next-byte algorithm in Section 4.2. Note that for $\bar{C}_{n+1}(a)$, we do not need to iterate over every time since the mapping from tokens to their prefix byte can be precomputed (line 30-33).

## B    Noteworthy implementation considerations

Theoretically, we do not have to predict into utf-8 byte space. We could predict into the largest set of tokens $\mathcal{A}$ that is contained in the vocabularies of all members of our model ensemble: $\max_{\mathcal{A}} \|\mathcal{A}\|$   s.t.   $A \subseteq \mathcal{V}_i$   for all $i$. This can have certain advantages, the primary one being that generation would be faster: for example imagine an average token has four bytes. Our method now needs to predict four times instead of one to generate the same expected amount of text. In this context, note that symbol based languages such as Chinese characters are often based on multiple bytes. Hence our method can be performance optimized by specializing the alphabet. Please note that our choice of alphabet does not change the statistical properties of the model, hence we would not expect accuracy differences due alphabet choice. However, the primary reason for us to choose the byte alphabet regardless was the ease of use for model ensemble: To run any experiment we need to map tokens probabilities to a **universal** alphabet, that is the easiest defining the alphabet as 255 bytes and the necessary control tokens.

## C    Ablation: Beam search

Each token is made of 6.6 byte on average. Thus, a tokenized model might have more foresight compared to a byte-level one when predicting one step at a time. Hence, even though, beam-search has not led to improvements in tokenized models (Holtzman et al., 2020; Stahlberg & Byrne, 2019; Koehn & Knowles, 2017), this might be different for byte-level models.

| Beam width | PSM | SPM |
|---|---|---|
| 1 | 65.7 | 63.9 |
| 2 | 66.8 | 65.9 |
| 4 | 67.7 | 66.8 |

Table 3: Performance increases for the next-byte prediction on our FIM-task when beam search is applied.

We repeat the FIM experiments with beam-width 1, 2 and 4, shown in Table 3. Consistently, we see an about 1% improvement for each prompting style and each beam-width increase. However, beam-search notably increases the memory footprint of any LLM because modern transformer based models need to store the KV-cache of each beam for efficient generation. Since the KV-cache makes about 2/3 of the overall memory consumption beam search is a memory intense operation.

**Algorithm 3** Compute $P_{\mathcal{A}}(.|x_1^n)$, which is a conditional distribution on $x_1^n$ over all bytes $x \in \mathcal{A}$. $C_{n+1}(a)$ and $\bar{C}_{n+1}(a)$ also contain the encoding probability $P(\vec{t})$ for each encoding they contain.

1: **procedure** COMPUTE($x_1^n$, cover_dict($x_1^n$) = None)
2:     **if** cover_dict($x_1^n$) = None:
3:         $P(x_1^n)$, cover_dict($x_1^n$) = EXTRACT_COVER($x_1^n$)
4:     *# Get the set $C_{n+1}(a)$ (See Section 4.2) for all $a \in \mathcal{A}$*
5:     $[C_{n+1}(a_1), ..., C_{n+1}(a_{|\mathcal{A}|})]$ = REGROUP_ENCODINGS(cover_dict($x_1^n$))
6:     *# Get the set $\bar{C}_{n+1}(a)$ (See Section 4.2) for all $a \in \mathcal{A}$*
7:     $\vec{t}$ = encode($x_1^n$); Query $P(\vec{t})$ from cover_dict($x_1^n$).
8:     $[\bar{C}_{n+1}(a_1), ..., \bar{C}_{n+1}(a_{|\mathcal{A}|})]$ =ADD_NEW_ENCODINGS($\vec{t}, P(\vec{t})$)
9:     *# Merge cover dictionaries.*
10:     **for** $a \in \mathcal{A}$ **do**
11:         $x_{n+1} = a$
12:         cover_dict($x_1^{n+1}$) = $C_{n+1}(a) \cup \bar{C}_{n+1}(a)$
13:         $P(x_1^{n+1}) = \sum_{\vec{t} \in \text{cover\_dict}(x_1^{n+1})} P(\vec{t})$
14:     **end for**
15:     Obtain $P_{\mathcal{A}}(.|x_1^n)$ by renormalizing $P(x_1^{n+1})$.
16:     **return** $P_{\mathcal{A}}(.|x_1^n)$, [cover_dict($x_1^{n+1}$) **for** $x_{n+1} = a \in \mathcal{A}$]
17: **end procedure**
18:
19: **procedure** REGROUP_ENCODINGS(cover_dict($x_1^n$))
20:     **for** $\vec{t}, P(\vec{t}) \in$ cover_dict($x_1^n$) **do**
21:         byte = decode($\vec{t}$)
22:         $C_{n+1}(\text{byte})[\vec{t}] = P(\vec{t})$
23:     **end for**
24:     **return** $[C_{n+1}(a_1), ..., C_{n+1}(a_{|\mathcal{A}|})]$
25: **end procedure**
26:
27: **procedure** ADD_NEW_ENCODINGS($t_1^k, P(t_1^k)$)
28:     Compute $P(t_{k+1} = t|t_1^k)$ for all $t \in \mathcal{V}$ *# Run 1 model inference for all $t$*
29:     Compute $P(t_1^{k+1}) = P(t_{k+1} = t|t_1^k)P(t_1^k)$ for all $t \in \mathcal{V}$ *# Broadcasting for all $t$*
30:     **for** $t \in \mathcal{V}$ **do**
31:         byte = decode($t_1^{k+1}$)
32:         $\bar{C}_{n+1}(\text{byte})[t_1^{k+1}] = P(t_1^{k+1})$
33:     **end for**
34:     **return** $[\bar{C}_{n+1}(a_1), ..., \bar{C}_{n+1}(a_{|\mathcal{A}|})]$
35: **end procedure**

## D   FIM CODE GENERATION EXAMPLES

Here, we show three examples from the random span FIM benchmark from Bavarian et al. (2022) that we evaluate on in Figure 6. Our greedy byte-level generations significantly improve upon the greedy token-level generations in the SPM (suffix-prefix-middle) mode with `CodeLlama-7b` because the prompt ends with the "prefix" portion of the code and the generation starts at a token boundary. Each figure visualizes an example from the benchmark with the prefix and suffix at the beginning and separated by `⟨FILL-ME⟩` indicating the portion for the LLM to generate. Then, we show the generations from the token and byte-level models in SPM mode along with token healing as a baseline. They illustrate exactly where tokenization issues arise when generating and show how the byte-level predictions correct them.

Listing 1: Code generation example with `CodeLlama-7b`. This one is interesting because of the typo in the variable "`delimeter`" in the dataset and the prefix ends with "`delim`". The misspelled "`delimeter`" is tokenized as two tokens: "`del imeter`" while the correctly spelled "`delimiter`" is tokenized as a single token. This makes the 1) token-level prediction incorrectly generate "`ter`" as the continuation because "`delimter`" is tokenized as three tokens "`del im ter`", which has a token boundary at the end of the prefix, 2) token-healed predictions incorrectly generate "`iter`" because at the token-level, "`delimiter`" is more likely, and 3) byte-level predictions correctly generate "`eter`" because the probability at the byte-level correctly marginalizes out the byte-level probability despite the context and many possible tokenizations.

```python
# RandomSpanInfilling/HumanEval/5/1
from typing import List

def intersperse(numbers: List[int], delimeter: int) -> List[int]:
    """ Insert a number 'delimeter' between every two consecutive
    ↪   elements of input list `numbers'
    >>> intersperse([], 4)
    []
    >>> intersperse([1, 2, 3], 4)
    [1, 4, 2, 4, 3]
    """
    if not numbers:
        return []

    result = []

    for n in numbers[:-1]:
        result.append(n)
        result.append(delim ⟨FILL-ME⟩ mbers[-1])

    return result
```

**token predictions** (did not pass ✗):
```
ter)
    result.append(nu
```

**token predictions with token healing** (did not pass ✗):
```
iter )
    result.append(nu
```

**byte predictions** (passed ✓):
```
eter)
    result.append(nu
```

Listing 2: Code generation example with `CodeLlama-7b`. This one is interesting because of the tokenization of "`collatz`" and the prefix starting with "`colla`". "`collatz`" is tokenized as two tokens "`coll atz`". This makes the 1) token-level prediction incorrectly generate "`zt`" as the continuation because "`collazt`" is tokenized as three tokens "`col la zt`", which has a token boundary at the end of the prefix, 2) token-healed predictions incorrectly generate "`pse`" because at the token-level after backing up, "`collapse`" (a single token) is more likely, and 3) byte-level predictions correctly generate "`tz`" because the probability at the byte-level correctly marginalizes out the byte-level probability despite the context and many possible tokenizations.

```python
# RandomSpanInfilling/HumanEval/123/1
def get_odd_collatz(n):
    """
    Given a positive integer n, return a sorted list that has the odd
    ↪   numbers in collatz sequence.

    The Collatz conjecture is a conjecture in mathematics that concerns a
    ↪   sequence defined
    as follows: start with any positive integer n. Then each term is
    ↪   obtained from the
```

```
    previous term as follows: if the previous term is even, the next term
    ↪   is one half of
    the previous term. If the previous term is odd, the next term is 3
    ↪   times the previous
    term plus 1. The conjecture is that no matter what value of n, the
    ↪   sequence will always reach 1.

    Note:
        1. Collatz(1) is [1].
        2. returned list sorted in increasing order.

    For example:
    get_odd_collatz(5) returns [1, 5] # The collatz sequence for 5 is [5,
    ↪   16, 8, 4, 2, 1], so the odd numbers are only 1, and 5.
    """
    if n%2==0:
        odd_collatz = []
    else:
        odd_colla ⟨FILL-ME⟩ 1

        if n%2 == 1:
            odd_collatz.append(int(n))

    return sorted(odd_collatz)
```

**token predictions** (did not pass ✗):
```
zt = [n]

    while n != 1:
        if n%2 == 0:
            n = n/2
        else:
            n = 3*n +
```

**token predictions with token healing** (did not pass ✗):
```
pse = []
        odd_collatz = [n]
        while n != 1:
            if n%2 == 0:
                n = n/2
            else:
                n = 3*n +
```

**byte predictions** (passed ✓):
```
tz = [n]

    while n != 1:
        if n%2 == 0:
            n = n/2
        else:
            n = 3*n +
```

Listing 3: Code generation example with `CodeLlama-7b`. This one is interesting because of the tokenization of "`palindrome`" and the prefix starting with "`palindr`". "`palindrome`" is tokenized as three tokens "`pal ind rome`". This makes the 1) token-level prediction incorrectly generate "`one`" as the continuation because "`palindrone`" is tokenized as four tokens "`pal ind r one`", which has a token boundary at the end of the prefix, 2) token-healed predictions incorrectly generate "`eome`" and results in the three-token chunk "`pal indre ome`" because the token healing did not successfully search backward enough, and 3) byte-level predictions correctly generate "`ome`" because the probability at the byte-level correctly marginalizes out the byte-level probability despite the context and many possible tokenizations.

```python
# RandomSpanInfilling/HumanEval/107/8
def even_odd_palindrome(n):
    """
    iven a positive integer n, return a tuple that has the number of even
    ↪  and odd
    integer palindromes that fall within the range(1, n), inclusive.

    Example 1:

        Input: 3
        Output: (1, 2)
        Explanation:
        Integer palindrome are 1, 2, 3. one of them is even, and two of
        ↪  them are odd.

    Example 2:

        Input: 12
        Output: (4, 6)
        Explanation:
        Integer palindrome are 1, 2, 3, 4, 5, 6, 7, 8, 9, 11. four of
        ↪  them are even, and 6 of them are odd.

    Note:
        1. 1 <= n <= 10^3
        2. returned tuple has the number of even and odd integer
        ↪  palindromes respectively.
    """
    def is_palindr ⟨FILL-ME⟩ odd_palindrome_count = 0

    for i in range(1, n+1):
        if i%2 == 1 and is_palindrome(i):
                odd_palindrome_count += 1
        elif i%2 == 0 and is_palindrome(i):
            even_palindrome_count += 1
    return (even_palindrome_count, odd_palindrome_count)
```

**token predictions** (did not pass ✗):

```python
one(n):
        return str(n) == str(n)[::-1]

    def is_palindrome(n):
        return is_palindrone(n) and n >= 10

    even_palindrome_count = 0
```

**token predictions with token healing** (did not pass ✗):

```python
eome(n):
        return str(n) == str(n)[::-1]

    even_palindrome_count = 0
```

**byte predictions** (passed ✓):

```python
ome(n):
        return str(n) == str(n)[::-1]

    even_palindrome_count = 0
```

# E  ANALYSIS OF TOKEN ALIGNMENT

Recently, Athiwaratkun et al. (2024) also proposed a similar method to mitigate the tokenization bias issue. In this method, they start regenerating the last $B$ tokens in the prompt by restricting the next-token to be either a prefix of the prompt's remaining string or cover it, i.e. the remaining string is the prefix of the generated token. Although this appears to overlap with our Corollary 1, the sampling procedure is biased and can produce invalid encodings due to the greedy masking process.

Consider the following example where we use the following vocabulary {`<_>`,`<1>`, `<a>`, `<_a>`, `<_aa>`} with MPE tokenizer. For a prompt string "`_a`" which is tokenized as `<_a>`, token alignment will regenerate from the beginning and force the generated tokens to match the template "`_a`". This is problematic when the first token generated is `<_>`. Given this token, tokenization bias occurs and the second token must not be `<a>` , since `<_a>` is a whole token. On the other hand, according to token alignment, the only possible next token after `<_>` is `<a>` , but `<_><a>` is an invalid encoding. The correct way to align is to by rejection sampling without greedy masking. We can use also our method by generating bytes until seeing a white space, then switch to token level generation. In this case, the prompt should exclude the last whitespace, and the first generated token must start with a whitespace.

