# OpenReview forum: "Exact Byte-Level Probabilities from Tokenized Language Models for FIM-Tasks and Model Ensembles"
_ICLR.cc/2025/Conference — ICLR 2025 Poster_

### Official Review · Reviewer_v6Xm · 2024-10-28

**Soundness:** 3
**Presentation:** 3
**Contribution:** 3
**Rating:** 6
**Confidence:** 4

**Summary:**

Tokenization is a common pre-processing step, which shortens input sequences by mapping multiple bytes to discrete tokens.
Prior work finds: Even for unlimited data and compute, tokenized models can achieve better cross-entropy loss vs. untokenized models
One hypothesis for why tokenization helps is that it allows models to handle longer contexts at less compute
Main problem: there are certain combinations that cannot be represented in token space, limiting the model’s abilities. This is harmful esp. for prompts that end mid-token.
Approach: this work presents a way to predict when tokenization is getting in the way and converts tokenized LMs into statistically equivalent byte-level models

**Strengths:**

- The characterization of the tokenization bias phenomenon, with examples / and the synthetic example with a 3rd order Markov chain, is very interesting and useful to the community
- The paper is well-written and studies an important problem

**Weaknesses:**

- The code infilling setting is an interesting example of where this work could be useful. However, it is not clear why the prefix would stop in the middle of a token/word or why new tokens would arise at inference time. Why are the <PRE>, <MID>, and <SUF> tokens needed? Why is the study restricted to two prompting formats/are others possible? Since tokenization schemes are lossless, it seems like any prefix should be representable with the set of available tokens at training time. Further explanation of the evaluation setting and why it is realistic would be helpful.
- Several prior methods can ensemble the predictions from models that do not share a vocabulary, or ensemble multiple predictions from a single model (https://arxiv.org/abs/2203.11171, https://arxiv.org/abs/2210.02441, https://arxiv.org/abs/2307.11031). It is not clear whether the ensembling method proposed in this paper would outperform these methods . The motivation that byte-representations help ensembling is not fully examined due to the lack of these comparisons.
- It would be useful to provide decoding efficiency metrics corresponding to each decoding approach in Figure 6 since efficiency is a motivation in Section 4.

**Questions:**

In the caption for Figure 2, should the probability of sampling “A” on the LHS be $(1 - \alpha)$? It states the probability is $\alpha$.

I would be willing to change my opinion based on the responses to the weaknesses!

---

> ### Author Response · Authors · 2024-11-20
> **Official Comment by Authors**
>
> Thank you for your questions! You can find our answers below
>
> **(1) The code infilling setting is an interesting example of where this work could be useful. However, it is not clear why the prefix would stop in the middle of a token/word or why new tokens would arise at inference time. Why are the <PRE>, <MID>, and <SUF> tokens needed? Why is the study restricted to two prompting formats/are others possible? Since tokenization schemes are lossless, it seems like any prefix should be representable with the set of available tokens at training time. Further explanation of the evaluation setting and why it is realistic would be helpful.**
>
> For context, before this work, tokenization bias has been observed empirically without clear theoretical explanation/understanding [1]. In scenarios such as code completion, this can be problematic as the model will output incorrect completion if the prompt ends with words that are partial tokens, especially the white space. A practical scenario is when the code developer stop typing at the middle of a word and the code LLM suggests wrong completion.
>
> Beside token healing and our method, one strategy to mitigate this issue is to finetune the model with PSM mode. The idea is that tokenization bias will not happen if the prompt ends with a control token (or equivalently- a synthetic control character that is not a part of any other token). In the case of PSM, the prompt always ends with the <MID> token for this reason.  The <PRE> and <SUF> tokens are inserted for the purpose of informing the model which part is a prefix/suffix that we want to complete. SPM format, however, is susceptible to tokenization bias since the prompt may end with a partial token. Nevertheless, SPM format provides a more natural flow and is more aligned with the pretrained data compared to PSM and for this reason, models are trained with both formats to maximize transfer between them (https://arxiv.org/pdf/2207.14255, section 8.1 and Appendix D; and https://arxiv.org/pdf/2308.12950, section 2.3).
>
> For these reasons, we test our method on the FIM benchmark as it allows us to rigorously compare its performance with other methods, such as PSM and token healing. Interestingly, with the byte level, SPM@10 performance is better and very comparable to PSM@10. Finally, we provide an example below about the PSM and SPM format.
>
> As an example, if we want to complete the **\<FILL_ME\>** in this code **“def add5(aa): r\<FILL_ME\> 5 + aa”**, then SPM format is:
>
> -  **\<PRE\>\<SUF\>** $\mathrm{encode}$("5 + aa") **\<MID\>**  $\mathrm{encode}$("def add5(aa): r") . Here the last token is \< r\> which incur tokenization bias.
>
> And the PSM format is:
>
> - **\<PRE\>** $\mathrm{encode}$("5 + aa")  **\<SUF\>** $\mathrm{encode}$("def add5(aa): r") **\<MID\>**. In this case, the completion is well separated by the **\<MID\>** token.
>
> [1] Dagan, G., Synnaeve, G., & Roziere, B. Getting the most out of your tokenizer for pre-training and domain adaptation. In Forty-first International Conference on Machine Learning.
>
> [2] Bavarian, M., Jun, H., Tezak, N., Schulman, J., McLeavey, C., Tworek, J., & Chen, M. (2022). Efficient training of language models to fill in the middle. arXiv preprint arXiv:2207.14255.

---

> ### Author Response · Authors · 2024-11-20
> **Official Comment by Authors (Cont)**
>
> **(2) Several prior methods can ensemble the predictions from models that do not share a vocabulary, or ensemble multiple predictions from a single model (https://arxiv.org/abs/2203.11171, https://arxiv.org/abs/2210.02441, https://arxiv.org/abs/2307.11031). It is not clear whether the ensembling method proposed in this paper would outperform these methods . The motivation that byte-representations help ensembling is not fully examined due to the lack of these comparisons.**
>
> Thank you for highlighting these relevant papers. We have reviewed them with great interest. Our initial observations are as follows:
> We believe that the ensemble techniques discussed in these papers are tailored to specific types of benchmarks. For instance, GSM8K requires a specific answering template, and the possible answers are typically limited to a small set (e.g., A, B, C, D), making voting a viable approach for these problems. To address your concern, we have extended our empirical results to include simple voting as well:
>
> | Dataset | Voting (all 3) | Voting (top 2) | Our |
> |---------|-------------------|-------------------|-----------------|
> | MMLU    | 0.593             | 0.621             | **0.654**           |
> | PIQA    | 0.791             | 0.801             | **0.807**           |
>
> Our method demonstrates its strengths in scenarios where voting is insufficient. This is why we included two coding benchmarks, which, to our understanding, cannot be effectively aggregated using the methods in [3]. Additionally, our ensemble method preserves the statistical properties of its members, potentially leading to more theoretical insights in the future.
> Moreover, our approach is flexible, as it only modifies the decoding process. This means that it is orthogonal to the methods you mentioned, and we could potentially combine them to enhance real-world results.
> We welcome any further thoughts you might have on this topic and are eager to refine our discussion further. Thank you for your effort in improving our paper!
>
>
> [3] Wang, Xuezhi, Jason Wei, Dale Schuurmans, Quoc Le, Ed Chi, Sharan Narang, Aakanksha Chowdhery, and Denny Zhou. "Self-consistency improves chain of thought reasoning in language models." arXiv preprint arXiv:2203.11171 (2022).
>
> [4] Arora, Simran, Avanika Narayan, Mayee F. Chen, Laurel Orr, Neel Guha, Kush Bhatia, Ines Chami, Frederic Sala, and Christopher Ré. "Ask me anything: A simple strategy for prompting language models." arXiv preprint arXiv:2210.02441 (2022).
>
> [5] Guha, Neel, Mayee Chen, Kush Bhatia, Azalia Mirhoseini, Frederic Sala, and Christopher Ré. "Embroid: Unsupervised prediction smoothing can improve few-shot classification." Advances in Neural Information Processing Systems 36 (2023): 63259-63291.
>
> **(3) It would be useful to provide decoding efficiency metrics corresponding to each decoding approach in Figure 6 since efficiency is a motivation in Section 4.**
>
> The PSM and SPM modes do not impact the efficiency of decoding because they are essentially methods for structuring prompts. On a single NVIDIA-A40, for CodeLlama2-7b, the generation speed at the byte level is approximately 0.14 seconds per byte, equating to about 0.8 seconds per token. However, when generating at the token level, the speed increases to roughly 0.08 seconds per token. This difference arises because each token typically contains about 7 bytes, and this is an intrinsic disparity between token-free and tokenized language models. For completeness, in byte-level decoding, the overhead computation (token merging, etc…) on average consumes 0.14-0.08=0.06s. Finally, we note that naively applying the BTR lemma by purely factorization without storing previously computed cover encodings likelihood would take roughly 4 minutes per byte.
>
> **(4) Typo in Figure 2.**
>
> Yes, it is indeed a typo. Thank you very much for pointing this out!
>
> --
> We appreciate your feedback. Let us know if additional points require clarification and we will try our best to accomodate!

---

### Official Review · Reviewer_KAHr · 2024-11-03

**Soundness:** 3
**Presentation:** 3
**Contribution:** 3
**Rating:** 6
**Confidence:** 4

**Summary:**

This paper provides a thorough investigation into the phenomenon of tokenization bias in language models, proposing a byte-level prediction method to mitigate this bias. The approach has promising implications, especially for tasks like fill-in-the-middle (FIM) in coding and for model ensembling. The idea of combining language models with distinct tokenizations at the logit level recalls older ensemble methods, commonly used before tokenization became mainstream in modern LLMs. The proposed method, underpinned by the Byte-Token Representation Lemma and efficient sampling algorithms, allows for effective ensemble models by bypassing tokenization restrictions, facilitating seamless integration across diverse language models.

**Strengths:**

What I really liked about this paper is that it very systematically investigates the phenomenon of tokenization bias. It develops a byte level prediction method and overcomes the bias. The applicability of this method and the findings in the paper have implications in areas such as coding where fill in the middle is crucial as well as in model ensembling. The idea of combining ensembles of foundation models at the logit layer is very appealing. Back in the day when there were no tokens and LMs were being built on whole words, ensembling using to be the norm in any practical LM based applications, such as speech recognition. With varying tokenization powering different LLMs today, the ability to freely combine any LLM with each other became restrictive. This paper proposes byte level efficient sampling algorithms and in conjuction with the byte-token representation lemma and the associated theoretical foundation, now allows the community to confidently combine predictions from any LLMs. Tokenization allows the LMs to overcome closed vocabulary limitation and from my perspective, this paper further allows us to overcome distinct vocabulary challenges associated with model ensembling.

**Weaknesses:**

While the results are promising for fill in the middle (in coding) tasks, I would have liked to see experimentation on other tasks to ascertain this method's generalizability to a wider variety of tasks, espically those that require context senstitive token predictions. Computer use, a new usecase demonstrated by companies such as Anthropic, could be a good use case that the authors could consider as an example.

An area that authors could delve further into is more analysis on the tokenization bias in multi step or long context tasks.

In order to truly appreciate the need for this research, its important to know just how much diversity there exists in tokenization vocabulary across different LLMs today.

**Questions:**

I am curious what do authors think about the generalizability of their methods to non-deterministic tokenization or in applications that involve mixed token vocabularies ? Could some experimental results be shown to further strengthen the paper ?

How do authors explain the not-so-strong gains with ensebling (Table 1 and Table 2). It was not clear how the models are being ensembled together? I would have also liked to see other combination strategies as a baseline e.g. consensus voting or ranking over diverse samples from different LMs. Are these experimental results available ?

In order to truly appreciate the need for this research, its important to know just how much diverseity there exists in tokenization vocabulary across different LLMs today. To that end, I would like to see some sort of distribution characterizing the popularity of tokenization and distribution of vocabulary sizes across various LLMs. If the world is already coverging to one tokenization and the vocabulary of tokens homogenizes over time, then what practical advantages does this paper bring ? We should debate whether or not this is true and I would like to see authors providing their rebuttal.

---

> ### Author Response · Authors · 2024-11-20
> **Official Comment by Authors**
>
> Thank you very much for your feedback and suggestions! Please find our response below
>
> **(1) While the results are promising for fill in the middle (in coding) tasks, I would have liked to see experimentation on other tasks to ascertain this method's generalizability to a wider variety of tasks, espically those that require context senstitive token predictions. Computer use, a new usecase demonstrated by companies such as Anthropic, could be a good use case that the authors could consider as an example.**
>
> Beside the FIM setting, our paper also shows the performance of byte-level predictions for different models, namely Llama2-7b, Yi-1.5b-6B, Mistral-7B-v0.3 on multiple reasoning tasks, namely MMLU, PIQA, NQ, TriviaQA and GSM8k, as shown in Table 1. Furthermore, other coding tasks are also included, namely code completion on Human Eval and MBPP, where we test on Codellama2 models and Yi-Coder-1.5B model. For each task, we compare the token-level and byte-level performance and show that they have similar performance. Note that in these tasks, tokenization bias does not seriously affect the performance since the prompt is well-structured. Finally, we highlight that the purpose of these tasks is to showcase the performance of ensembling byte-level models.
>
> Thank you for your suggestion on the computer-use, which looks very interesting! This method, however, was recently introduced and we are not aware of any benchmark/task available for us to provide a concrete evaluation. The reason we choose the FIM tasks is because it allows us to benchmark and compare against other existing fixes, namely PSM mode and token-healing.
>
> **(2) An area that authors could delve further into is more analysis on the tokenization bias in multi step or long context tasks.**
>
> This is an interesting point! We think it relates to many practical aspects in LLM sampling. Our first intuition is, when LLMs predict the next token, any bias introduced in this process can compound over long sequences. If the model has a tendency to favor certain tokens over others due to its training data or the tokenization method, this preference can lead to a skewed prediction trajectory. This can be significant for mathematical/symbolic reasoning tasks, since preferring one token over another can potentially lead to a drift from the intended meaning or logical path. For example, suppose we ask the LLM **“what is 11+11=”** and **<22>** is a token in the vocabulary. If the LLM samples the first token as **<2>** then due to tokenization bias, it must output another token that is not **<2>**. Also see [1]. Overall, we believe this is an exciting direction that requires extensive analysis for future work.
>
> [1] Dagan, G., Synnaeve, G., & Roziere, B. Getting the most out of your tokenizer for pre-training and domain adaptation. In Forty-first International Conference on Machine Learning.
>
> **(3) I am curious what do authors think about the generalizability of their methods to non-deterministic tokenization or in applications that involve mixed token vocabularies ? Could some experimental results be shown to further strengthen the paper ?**
>
> Thank you for your suggestion. We think this is nice and will help improve our paper’s result. See (1) and (2) in general response for details. Short answer: some non-deterministic tokenization was introduced to counter the tokenization bias problem, such as BPE Dropout. This is equivalent to training the LLM with different vocabulary sets (but still using the BPE token merge rule). Theoretically, BPE-dropout can still incur tokenization bias, we illustrate this by using the same 1st order Markov chain example in the paper. It turns out that the Byte-Token representation lemma works for this case as well.

---

> ### Author Response · Authors · 2024-11-20
> **Official Comment by Authors (Cont)**
>
> **(4) How do authors explain the not-so-strong gains with ensebling (Table 1 and Table 2). It was not clear how the models are being ensembled together? I would have also liked to see other combination strategies as a baseline e.g. consensus voting or ranking over diverse samples from different LMs. Are these experimental results available ?**
>
> Model ensemble: we first convert each model to their statistical byte-level counterpart. Given a prompt, each model will output a distribution over all possible next bytes, which we ensemble them by averaging their predictive distributions. Then, we sample a new byte according to this averaged distribution and repeat the autoregressive process.
>
> Ensemble Performance: while ensembling improves the performance in general, it depends on the performance of each individual model and many other factors, such as training data, domains, calibration, etc. In fact, we believe a more optimal way to ensemble models is weighted averaging, where the weights are determined by the input, similar to the mixture of expert schemes. Overall, the performance boost by ensemble is an interesting topic that requires systematic study.
>
> We are also happy to report the result for the voting benchmark with tie-breaking on multiple choice tasks, in this case MMLU and PIQA. For multiple-choice tasks, given a set of statements as a string, each LLM will output their answer by ranking the likelihood of each statement. Majority voting will output the selection that is commonly selected among the LLMs.
>
> | Dataset | Voting (all 3) | Voting (top 2) | Our |
> |---------|-------------------|-------------------|-----------------|
> | MMLU    | 0.593             | 0.621             | **0.654**           |
> | PIQA    | 0.791             | 0.801             | **0.807**           |
>
> Here, our ensemble method outperforms voting. Finally, we note that voting is not   trivially extended to reasoning tasks such as code completion without training an evaluator network, as discussed in the Related Work section.
>
>
> **(5) In order to truly appreciate the need for this research, its important to know just how much diverseity there exists in tokenization vocabulary across different LLMs today. To that end, I would like to see some sort of distribution characterizing the popularity of tokenization and distribution of vocabulary sizes across various LLMs. If the world is already coverging to one tokenization and the vocabulary of tokens homogenizes over time, then what practical advantages does this paper bring ? We should debate whether or not this is true and I would like to see authors providing their rebuttal.**
>
> Absolutely! Thank you for highlighting the importance. See our general response (4).
>
> --
>
> We appreciate your feedback! Let us know if you have further questions!

---

> > ### Comment · Reviewer_KAHr · 2024-12-02
> >
> > Thank you for your response on model ensembling question. I dont, however, think, I got the full insight into token vocabulary homogenization point i was trying to make in \5.

---

> ### Author Response · Authors · 2024-12-02
>
> Thank your for your response!
>
> Happy to elaborate on the question of understanding token diversity.
>
> (1) In our first response, we wanted to highlight the **theoretically optimal case**. We argued that the vocabulary will be significantly different depending on the model size, and the vocabulary will strongly depend on the task at hand [1,2].
> (2) Practically, speaking to the **differences in vocabularies today's models**, we pointed to a study that showed a table in how much different tokenizer vocabularies influence the compression of different languages hence pointing to significant differences in today's models [3]. We want to augment this argument further and derived the following statistics for the models we used in our study:
>
> | Model      | Vocab Size | # Tokens in Common | % of Common Tokens | # of Unique Tokens | % of Unique Tokens |
> |------------|------------|--------------------|--------------------|--------------------|--------------------|
> | LLama2-7B      | 32,000     | 18,838             | 58.9%              | 6,353              | 19.85%             |
> | Yi-1.5-6B      | 63,992     | 18,838             | 29.4%              | 38,169             | 59.65%             |
> | Mistral-7B-v0.3 | 32,768     | 18,838             | 57.5%              | 3,062              | 9.34%              |
>
> As you can see for each model the number of tokens all models have in common is not close to 100% of the vocabulary size for any model, and the number of unique tokens in each model take a significant proportion. We would suspect the differences to be even larger for more specific models such as language or task specific models. We hope this underlines the relevance of our method.
>
> Please let us know if this is what you had in mind. Happy to continue the discussion, excited to hear your opinion :)

---

### Official Review · Reviewer_X7br · 2024-11-04

**Soundness:** 3
**Presentation:** 2
**Contribution:** 2
**Rating:** 5
**Confidence:** 4

**Summary:**

This paper investigates the impact of tokenization on the performance of language models, particularly highlighting a phenomenon called "tokenization bias." This bias comes from the discrepancies between the predictive distributions of tokenized LMs and their statistically equivalent byte-level distribution when predicting the next byte or token. In this paper, they introduce the Byte-Token Representation Lemma (BTR Lemma) to map between the token and byte domain, which allows the computation of exact byte-level probabilities without requiring extra modification of the model. This method identifies "cover encodings" for a given byte string which are all possible valid tokenizations of a string that "optimally" contain the string in the prompt. Experiments on FIM code completion task and model ensemble task show that the proposed method improves model performance.

**Strengths:**

* This paper defines the tokenization bias phenomenon, which could cause performance issues in practical scenarios.
* They evaluated the proposed method on FIM code completion task and mode ensemble task.
* The proposed byte-level prediction method considers both the accuracy and efficiency in recovering the unbiased byte-level distribution.

**Weaknesses:**

* This method is very similar to the ACL 2024 paper 'Token Alignment via Character Matching for Subword Completion' which addresses the same issue and proposes more evaluation scenarios. However, in this paper, they didn’t compare with the token alignment method.

* Based on the results shown in Figure 6, the improvement of the proposed method on the FIM evaluation tasks in sampling settings (pass@10) over the token-level with token healing method is very minor. For example, from 84.0 to 84.1. Why is that?

* The evaluation scenarios are limited. Only two types of tasks are covered. How’s the performance of the method on general completion tasks without the FIM setting? Does the tokenization bias also have an impact on natural language completion tasks?

**Questions:**

* How is the proposed method compatible with other tokenization algorithms besides BPE and MPE?
* What do the right two sub-figures mean in Figure 6?

---

> ### Author Response · Authors · 2024-11-20
> **Official Comment by Authors**
>
> Thank you very much for your helpful feedback on the paper! You can find your concerns addressed in the following
>
> **(1) This method is very similar to the ACL 2024 paper 'Token Alignment via Character Matching for Subword Completion' which addresses the same issue and proposes more evaluation scenarios. However, in this paper, they didn’t compare with the token alignment method.**
>
> Thank you for bringing this ACL 2024 work to our attention—we were not previously aware of it! You can find the detailed discussion in General Response (3). In short, token-alignment sampling has inherent sampling biases and may produce invalid encodings, which fall outside the token domain distribution. Nevertheless, the approach has conceptual overlap with our method, i.e. Corollary 1.
>
> **(2) Based on the results shown in Figure 6, the improvement of the proposed method on the FIM evaluation tasks in sampling settings (pass@10) over the token-level with token healing method is very minor. For example, from 84.0 to 84.1. Why is that?**
>
> We would like to clarify that for the PSM mode, the token and byte-level mode have very similar performance. In this mode, a control token <MID> is deliberately inserted after the prompt. This is a specific training technique developed to mitigate the tokenization bias in practice (designed for coding tasks), which however, requires further training. The fact the byte-level achieves similar performance compared to token-level is expected in this case, i.e. @1: 64.9 (token) versus 65.7 (byte) and @10: 84.0 (token) versus 84.1 (byte), since the two models are statistically equivalent in this case.
>
> In the SPM mode, we observe that byte-level outperforms token level and token-healing. The poor performance of token-level in SPM mode is due to tokenization bias, since the prompt can end with a partial token. Since token-healing is a technique to mitigate this issue, the performance gap between byte-level prediction and token-level (+ token healing) is smaller. Furthermore, we note that there are scenarios where token healing is unable to fix the bias, as shown in Appendix D, since it assumes that the issue can be fixed with 1 single token that covers the whole string. As an example, consider the setting in Figure 4 with another query string “$\texttt{yummyu}$”, which is tokenized into $\texttt{<yum>}$ and $\texttt{<myu>}$. Prompting to the LLM, it will not sample the token $\texttt{<m>}$ due to tokenization bias.  This cannot be fixed with token healing, since $\texttt{<myu>}$ is already the longest token that contains $\texttt{“myu”}$. Our algorithm correctly fixes this by including the correct cover encoding for this scenario, which is $\texttt{<yum>, <m>}$, and $\texttt{<yum>}$.
>
> **(3) The evaluation scenarios are limited. Only two types of tasks are covered. How’s the performance of the method on general completion tasks without the FIM setting? Does the tokenization bias also have an impact on natural language completion tasks?**
>
> In addition to the FIM setting, our paper also evaluates the performance of byte-level predictions across various models, including Llama2-7B, Yi-1.5B-6B, and Mistral-7B-v0.3, on a range of reasoning and knowledge tasks such as MMLU, PIQA, NQ, TriviaQA, and GSM8k, as detailed in Table 1. We also include results for coding tasks, specifically code completion on HumanEval and MBPP, using CodeLlama2 models and Yi-Coder-1.5B models. If there are any specific benchmarks you would like us to explore further, please feel free to let us know—we would be happy to consider them.
>
> For each task, we compare the performance of token-level and byte-level approaches, demonstrating that they achieve similar results. This is because, in these tasks, tokenization bias has minimal impact due to the well-structured nature of the prompts.
> Importantly, the purpose of these evaluations is to showcase the performance improvements achieved by ensembling byte-level models. Without requiring any additional training, this approach yields consistent gains across nearly all tasks, which we find particularly compelling. While ensembling reduces bias and variance, its effectiveness ultimately depends on the underlying members. That said, achieving gains of approximately 4% on tasks such as coding highlights the potential of this method.
> If there are specific tasks you would like us to explore, we would be happy to accommodate your suggestions and provide additional results.

---

> ### Author Response · Authors · 2024-11-20
> **Official Comment by Authors (Cont)**
>
> **(4) How is the proposed method compatible with other tokenization algorithms besides BPE and MPE?**
>
> See the general response (1) and (2)
>
> **(5) What do the right two sub-figures mean in Figure 6?**
>
> The right two sub-figures in Figure 6 show the effects of temperature scaling on the performance of the byte-level and token-level models. This is necessary since unlike greedy decoding, temperature scaling has a significant impact on the model’s performance in the @10 benchmark. Specifically, high temperature encourages models to explore a more diverse solutions while low temperature will force the model to narrow down close to greedy sampling. In this case, for each model, we choose the temperature that yields the best performance for both token and byte-level model.
>
>
> Let us know if our reply address your concern! Thank you very much for the feedback!

---

> ### Comment · Reviewer_X7br · 2024-12-02
> **Response to Authors**
>
> Thanks for providing the clarifications. However, the example in general response (3) looks confusing to me.
> >On the other hand, according to token alignment, the only possible token is <a> but <_><a> is an invalid encoding.
>
> Given token alignment is doing backtracking via byte level trie, why is the only possible token <a>?

---

> ### Author Response · Authors · 2024-12-02
>
> Hi, thank you for your response, we really appreciate it! You can find our detail clarification below.
>
> For convenience, we recite the vocabulary in the example here **{<_>, <1>, <a>, <_a>, <_aa>}**. The prompt we are considering is **"_a"**, which is tokenized as **<_a>**. Here, tokens are within the bracket **<>**.
>
> In this example, since the prompt is exactly 1 token, i.e. **<_a>**, we backtrack 1 token which is equivalent to regenerating the encodings that have prefix **"_a"** (i.e. cover encodings in our paper). For the first token, token alignment requires it to either:
>
> (1)  Be a prefix of a string **"_a"**: In this case, we have two candidates, which are **<_>** and **<_a>**.
>
> (2) **"_a"** must be the prefix of this token: In this case, we have two candidates **<_a>** and **<_aa>**.
>
> So in total, for the first token, we have 3 candidates **<_>**, **<_a>** and **<_aa>**. Suppose the model sampled **<_>**, then the next token, i.e. the second token, to be generated must:
>
> (1) Be a prefix of the remaining string **"a"**: the candidate is now **<a>** since it is the only token that matches this criteria.
>
> (2) **"a"** must be the prefix of this token: the only candidate for this case is again **<a>**.
>
> Hence, for the second token after **<_>** has been generated, the only viable option according to token-alignment is **<a>**. As such, token-alignment will force the model to sample **<a>**. But now the first and second token form an invalid encoding that is **<_><a>** which has zero probability of occurrence in the token-level distribution.
>
> Thank you for your question and feel free to let us know if you have any further feedback!

---

### Official Review · Reviewer_oJBc · 2024-11-04

**Soundness:** 3
**Presentation:** 3
**Contribution:** 3
**Rating:** 8
**Confidence:** 3

**Summary:**

This paper reveals a fundamental issue in language models regarding tokenization bias - where tokenized and byte-level models, despite being statistically equivalent, produce different predictive distributions. The authors introduce the Byte-Token Representation Lemma, providing a framework to convert tokenized language models into statistically equivalent token-free ones without additional training. They demonstrate practical applications in fill-in-the-middle tasks and model ensembles, achieving significant improvements: 18% on FIM coding benchmarks and up to 3.7% on ensemble tasks. The paper provides both theoretical foundations and practical implementations through efficient algorithms for cover encoding search and next-byte sampling.

**Strengths:**

The authors attempted to address a critical problem on tokenization for LLMs. They are the first to introduce tokenization bias and its effects on model behavior. They present a theory of it and introeduced efficient implementation of the algorithm.

The authors also studied the practical impact of the proposed solution through both code generation task (FIM, speficially) and model emsambles. Results indicate that this training-free approach is working well.

**Weaknesses:**

While the current results are great, it'd be better if authors can also cover tasks other than FIM that suffer from this issue, e.g. complete a story in natural language or generic code completion task that doesn't require FIM.

The authors discussed computational overhead but doesn't thoroughly analyze the latency impact especially in a practical setting. It'd be better if authors can study these as part of the experiments authors conducted.

The interaction between the proposed byte-level prediction method and popular sampling techniques (like  top-p, top-k) have not been explored enough, especially in the ensemble context.

**Questions:**

See weakness above. In addition,

1. Do authors plan to open-source the implementation?

2. There is also a line of work similar to token healing and seems working well in both code and text. Could the authors study it and see whether the token alignment method could mitigate the issue especially in single-model completion tasks?
Token Alignment via Character Matching for Subword Completion https://aclanthology.org/2024.findings-acl.929.pdf

---

> ### Author Response · Authors · 2024-11-20
> **Official Comment by Authors**
>
> Thank you for your thoughtful comments—here are some clarifications addressing your points.
>
> **(1) While the current results are great, it'd be better if authors can also cover tasks other than FIM that suffer from this issue, e.g. complete a story in natural language or generic code completion task that doesn't require FIM.**
>
> Thank you for your suggestion! We acknowledge that tasks like story completion and generic code completion are also relevant. However, story completion benchmarks often rely on human evaluations, making it challenging to obtain standardized comparisons (please feel free to correct us if we are mistaken). We chose the FIM task as it enables a direct comparison with existing methods that address tokenization bias in a quantifiable manner.
>
> For generic code completion tasks such as HumanEval and MBPP, we observed that performance differences between byte-level and token-level approaches are minimal. This is because the prompts in these benchmarks are typically well-structured and predominantly consist of requirement descriptions, which mitigate tokenization bias.
>
> That said, if you have a specific benchmark in mind, we would be happy to explore it and incorporate the results into our study.
>
> **(2) The authors discussed computational overhead but doesn't thoroughly analyze the latency impact especially in a practical setting. It'd be better if authors can study these as part of the experiments authors conducted.**
>
> Compared to the token-level generation, byte-level generation is roughly 7 times slower due to the overhead (also each token contains multiple-bytes). This can be overcome nevertheless, by using byte-level generation to fix the tokenization bias and then return to the token-level generation. For model ensembles, however, byte level generation is necessary for ensembles. Nevertheless, we believe that there is improvement potential that one can develop to make this more efficient.
>
> For context, on a single NVIDIA-A40, with CodeLlama2-7b, the byte-level generation speed is roughly 0.14 seconds per byte, which translates to about 0.8 seconds per token. In contrast, at the token level, generation speed is approximately 0.08 seconds per token. This variation occurs because, on average, a token contains around 7 bytes, highlighting the fundamental decoding speed difference between token-free and tokenized language models. Regarding computational overhead, in byte-level decoding, processes like token merging add about (0.14s - 0.08s) = 0.06 seconds to the process. Lastly, without retaining previously calculated cover encodings' likelihoods, simply applying the BTR lemma through factorization would consume approximately 4 minutes per byte.
>
> **(3) The interaction between the proposed byte-level prediction method and popular sampling techniques (like top-p, top-k) have not been explored enough, especially in the ensemble context.**
>
> This is a good point. To clarify, in our experiments, we do not use top-p sampling for byte-level generation. For token-level generation, we found using the parameters suggested by the model creator yields the best results, e.g. top_p=0.9. Nevertheless, the results for both cases in Table 1 and 2 are very similar. For systematic studying the impact of top-p and top-k on text degeneration, as in [1], we defer this as future work. Let us know if this addresses your question.
>
> [1] Holtzman, Ari, Jan Buys, Li Du, Maxwell Forbes, and Yejin Choi. "The Curious Case of Neural Text Degeneration." In International Conference on Learning Representations.
>
> **(4) Do authors plan to open-source the implementation?**
>
> Yes, the implementation will be open-source for reproducibility, using HuggingFace framework.
>
> **(5) There is also a line of work similar to token healing and seems working well in both code and text. Could the authors study it and see whether the token alignment method could mitigate the issue especially in single-model completion tasks? Token Alignment via Character Matching for Subword Completion https://aclanthology.org/2024.findings-acl.929.pdf**
>
> Thank you for sharing this ACL-2024 work which we are not aware of! Please see details in general response (3). In short, token-alignment sampling is related to our Corollary 1 but the sampling process is biased and can produce invalid encodings, which is out-of-distribution in the token domain.
>
> -
>
> We greatly appreciate your comment on the paper. Let us know if any further clarifications required

---

> > ### Comment · Reviewer_oJBc · 2024-12-01
> >
> > I appreciate authors' response. I am keeping my score assuming that authors will add discussion point (2) into the paper and commit to point (4).

---

> ### Author Response · Authors · 2024-12-01
>
> Thank you for your replies, we really appreciate it! We will add the discussion and release the code. Also, we answer your questions in the general response above. Let us know if you need any further clarification.

---

### Author Response · Authors · 2024-11-20
**General Response**

We thank the reviewers for your constructive feedback on our work, we’re excited to improve the draft further together. Multiple reviewers enjoy the insights our work provided as well as suggest many interesting directions which improves our work. Here, we provide a response to some general questions in the review.

**(1) Other tokenization schemes [Reviewer X7br, KAHr]:** Most common available tokenizers are using either BPE (Sentencepiece, tiktoken) or MPE (WordPiece). Our Byte-Token Representation Lemma applies to all of the aforementioned, and more specifically any deterministic tokenizer, as indicated in Section 4.1. The theoretical result that specifically applies to BPE/MPE is Proposition 1 in Section 3.2.2, i.e. invalid encodings do not occur in the token-level distribution. Leveraging this result, we can efficiently perform byte-level sampling, as shown in Section 4.2. Finally, we believe that our proof for tokenization bias (Proposition 1, Appendix A4) can be extended to future tokenization algorithms.

**(2) New Results for Non-deterministic Tokenizer [Reviewer X7br, KAHr]:** Another class of tokenizer is non-deterministic, such as BPE Dropout, to combat model’s weakness against text fragmentation by randomly omitting tokens before text processing, theoretically training the model across multiple vocabularies. Although intended to enhance robustness, our analysis suggests that tokenization bias still exists. It is partially mitigated, nevertheless, since the model needs to minimize loss across varied vocabulary sets. The BTR lemma (Lemma 1) holds, requiring consideration of all valid encodings under this variability. In practice, given the substantial size of the vocabulary, BPE-dropout is expected to be robust to tokenization bias, i.e.assigning non-zero probabilities to invalid encodings. Yet, it is not commonly employed in training large-scale LLMs because the randomized tokenization introduces a computational bottleneck, which slows down the training process. Also, it may require greater model capacity to handle the increased complexity.


For experiments, we use the 1st order Markov chain example in the paper where $(\alpha, \beta)= (0.4,0.3)$. We train a LLM with BPE dropout where a sequence can be tokenized either with the vocabulary {$\texttt{<A>,<B>,<AA>\}$} or {$\texttt{<A>,<B>\}$}. Consider the following  input tokens and their next-token probabilities:

- $\texttt{``<B> <A>”}$ and {$\texttt{<A>}: 0.43, \texttt{<B>}: 0.57, \texttt{<AA>}: 10^{-5}$}. tokenization bias happens according to Def 1 but the probability does not go to 0.0 as the cross entropy loss is distributed between the two vocabularies.

- $\texttt{``<B> <A><A>”}$ and {$\texttt{<A>}: 0.59, \texttt{<B>}: 0.41, \texttt{<AA>}: 10^{-5}$} which approximately equal to the original byte-level probabilities since the LLM detects that the only possible vocabulary that produces these tokens is {$\texttt{A,B}$}.

- $\texttt{``<B> <AA><A>”}$ and {$\texttt{<A>}: 10^{-4}, \texttt{<B>}: 0.99, \texttt{<AA>}: 10^{-5}$}: **tokenization bias occurs**, i.e. the LLM only outputs $\texttt{<B>}$. Since $\texttt{<AA>}$ only occurs in the vocabulary {$\texttt{<A>,<B>,<AA>\}$}, **the LLM detects the tokenization pattern** and infer that the sequence must be tokenized with this vocabulary.

Finally, we apply the BTR Lemma to remove the bias for the string $\texttt{``BA''}$ where $P(\texttt{<B><A>})=0.104$, $P(\texttt{<B><AA>})=0.0415$. Using $\texttt{BAA}$, we have $P(\texttt{<B><A><A>})=0.0443$ and $P(\texttt{<B><AA>})=0.0415$, which gives us
 $\alpha=0.405$ after factorization.

---

> ### Author Response · Authors · 2024-11-20
> **General Response (Continue)**
>
> **(3) Token-Alignment  https://aclanthology.org/2024.findings-acl.929.pdf [Reviewer oJBc,X7br]**: While token-alignment (TA) can be considered a special case of our Corollary 1, the sampling procedure is biased and can produce invalid encodings due to the greedy masking process. Consider this example where we use a vocabulary { $\texttt{<}$\_$\texttt{>}$, $\texttt{<1>}$, $\texttt{<a>}$, $\texttt{<}$\_$\texttt{a}$$\texttt{>}$, $\texttt{<}$\_$\texttt{aa}$$\texttt{>}$ } with MPE tokenizer. For the prompt string "_$\texttt{a}$", which is tokenized as $\texttt{<}$_$\texttt{a}$$\texttt{>}$, TA will regenerate from the beginning and force the generated tokens to match the template $\texttt{``}$\_$\texttt{a''}$. This is problematic when the first token generated is $\texttt{<}$_$\texttt{>}$. Given this token, tokenization bias occurs and the second token must not be $\texttt{<a>}$, since $\texttt{<}$_$\texttt{a}$$\texttt{>}$ is a token. On the other hand, according to token alignment, the only possible token is $\texttt{<a>}$ but $\texttt{<}$_$\texttt{><a>}$ is an invalid encoding. The correct way to align is to by rejection sampling without greedy masking. We can use also our method by generating bytes until seeing a white space, then switch to token level generation. In this case, the prompt should exclude the last whitespace, and the first generated token must start with a whitespace.
>
> We also believe our work goes beyond token alignment by introducing a method to convert tokenized LLMs into byte-level counterparts, enabling exact likelihood computation and effective ensembling. Additionally, we provide a clear probabilistic framework to address tokenization bias, offering a principled alternative to existing approaches.
>
> **(4) On diversity in tokenization vocabulary across different LLMs today and in the near future [Reviewer KAHr]:** This is absolutely an important point. Here are two key findings from the tokenization literature that underscore this diversity, and we think underlines the importance of our research.
>
> - Model Size and Vocabulary Size Correlation: Prior work has shown that the optimal vocabulary size is closely linked to the model size, both empirically and theoretically [1]. Specifically, it has been demonstrated that the optimal vocabulary sizes for models of varying sizes often differ by a factor of n > 1 (refer to Figure 2 in the cited paper). This implies that vocabularies for smaller models are at least subsets of those for larger models.
>
> - Domain-Specific Vocabulary Needs: Beyond model size, the optimal tokenization strategy can vary significantly based on the specific use case or domain. As noted in [2], achieving optimal cross-entropy loss in transformers is contingent upon selecting an appropriate tokenization approach. Furthermore, [3] provides a comparative analysis of different tokenizers and languages in their appendix, illustrating that different applications and languages may require distinct tokenization strategies to achieve optimal performance.
>
> - The Size of Universal Vocabulary: We note that the size of all utf8 characters is roughly 1 million (https://en.wikipedia.org/wiki/UTF-8). Given this, constructing a universal token vocabulary capable of encompassing all languages and characters would be of considerable magnitude, potentially leading to significant resource demands (for reference, llama2 vocabulary size is 32k, which contains mostly English tokens).
> These findings emphasize the necessity for tailored tokenization approaches to accommodate the diverse requirements of various LLMs and their applications.
>
> [1] Tao, Chaofan, Qian Liu, Longxu Dou, Niklas Muennighoff, Zhongwei Wan, Ping Luo, Min Lin, and Ngai Wong. "Scaling laws with vocabulary: Larger models deserve larger vocabularies." arXiv preprint arXiv:2407.13623 (2024).
>
> [2] Rajaraman, Nived, Jiantao Jiao, and Kannan Ramchandran. "Toward a Theory of Tokenization in LLMs." arXiv preprint arXiv:2404.08335 (2024).
>
> [3] Rahman, Abrar, Garry Bowlin, Binit Mohanty, and Sean McGunigal. "Towards Linguistically-Aware and Language-Independent Tokenization for Large Language Models (LLMs)." arXiv preprint arXiv:2410.03568 (2024).

---

> > ### Comment · Reviewer_oJBc · 2024-12-01
> >
> > Thanks for summary.
> >
> > On (3), could you clarify what you meant by "rejection sampling without greedy masking"?
> >
> > On (4), would you expect a performance difference when you merge two models with similar vocab or more distinct vocab?

---

> > > ### Author Response · Authors · 2024-12-01
> > > **Reply to Reviewer oJBc**
> > >
> > > Hi, thank you for your questions. We clarify them below.
> > >
> > > 1) On "rejection sampling without greedy masking": this method involves generating the subsequent tokens randomly, without the strict constraint of token alignment. Specifically, if these randomly generated tokens do not match the desired suffix string, we discard them and repeat the process until we find a sequence that does match. This will ensure that we do not accept invalid encoding as in the example shown above.
> > >
> > > 2) We believe having some overlap is favorable because it suggests that the models have a common ground in their understanding of tasks or domains. Conversely, for instance, if models are trained on completely different languages, resulting in unique vocabularies, this might reduce the effectiveness of ensemble methods due to a lack of shared context understanding [1].
> > >
> > > Let us know if you need further clarifications!
> > >
> > > [1] Dietterich, Thomas G. "Ensemble methods in machine learning." International workshop on multiple classifier systems. Berlin, Heidelberg: Springer Berlin Heidelberg, 2000.

---

> > > > ### Comment · Reviewer_oJBc · 2024-12-01
> > > >
> > > > Thanks for the response.
> > > >
> > > > On 1, could we apply the same method in TA by sampling a lot and dropping invalid ones? Revisiting the TA paper and it appears they also used sampling despite not dropping invalid encoding explicitly (which IMO might or might not be needed as invalid encoding will likely have high ppl and thus could still be filtered out)?
> > > >
> > > > On 2, "Conversely, for instance, if models are trained on completely different languages, resulting in unique vocabularies, this might reduce the effectiveness of ensemble methods due to a lack of shared context understanding" wouldn't this actually be the excat use cases that require ensembling? For example, considering conbining speciality models trained on English vs Hindu vs Japanese, or text vs math vs code?

---

> ### Author Response · Authors · 2024-12-01
> **Reply to Reviewer oJBc**
>
> Thank you for your insights!
>
> (1) We believe your proposal is promising. Interestingly, it shares some similarities with our Corollary 1, in which we pinpoint all valid cover encodings starting with designated prefix tokens. So indeed, we can compute the next valid token probability by applying your proposal with our cover encodings extraction method! Finally, we believe filtering invalid encodings is necessary because the greedy masking operator used in TA methods can force the model to accept invalid tokens that otherwise would have no chance (or with very low probability) of being selected, as demonstrated in our example in the original response. In our experiment with TA, we observe that this often happen when the number of regenerated tokens are large (e.g. we regenerate from the beginning of the prompt). In general, we believe our framework provide improvement insights to TA.
>
> (2) Thank you for your insights! In our previous response, we focus on the performance enhancements demonstrated in the paper, noting that when models show significantly different results on certain tasks, as seen in GSM8K (Table 1), a uniform averaging of predictions might not yield the best results. For this reason, the consideration of overlapping or distinct vocabularies was mainly approached as a heusristic due to our lack of detailed knowledge about the training data for each LLM.
>
> When it comes to leveraging the strengths of multiple LLMs, we think adaptive approach like Mixture-of-Experts, where the predictions are weighted dynamically to optimize the ensemble output, would yield the best result. For example, in case of text+math+code, we speculate that the prediction weighted towards more from math models when answering math questions . Overall, we believe that our technique of converting tokenized models to operate at the byte level could serve as a valuable tool for such research.
>
> Feel free to let us know if you need further clarification!

---

### Author Response · Authors · 2024-11-25
**Encouragement to discuss + additional experimental results**

The discussion period will be closing soon, we hence encourage all reviewers to respond to our rebuttal. We highly value your feedback and would appreciate your contributions to the discussion. Please take this opportunity to share your perspectives and engage with the community. Thank you for your participation and support!


Additionally we would also like to share some additional experimental work, we implemented token alignment as an additional baseline for the FIM task.

| Method | SPM (Greedy @1) | SPM (@10)  |
|---------|-------------------|-------------------|
| Token Alignment   | 63.0             | 82.9             |
| **Our**   | **63.9**             | **84.3**            |

Both results lack behind our method.

---

### Author Response · Authors · 2024-12-04
**Final comment before discussion period ends | Summary of rebuttal contributions**

Reviewers,

Thank you for your feedback on our submission. We have made several improvements and clarifications based on your comments.

***Summary of Additional Contributions and Clarifications:***

**1. New Experimental Results**: Introduced comparisons with token alignment as a baseline for the FIM task, showing our method's superior performance.

**2. Theoretical Insights on Stochastic Tokenizers**: Expanded our framework to include non-deterministic tokenizers like BPE Dropout, demonstrating partial mitigation of tokenization bias.

**3. Clarifications on Token-Alignment**: Explained limitations of token-alignment methods and highlighted our approach's ability to enable exact likelihood computation and effective ensembling.

**4. Discussion on Tokenization Diversity**: Provided insights into the diversity of tokenization vocabularies across models, underscoring the practical importance of our research.

**5. Ensemble Method Comparisons**: Demonstrated our method's superior performance over simple voting strategies, especially in complex tasks like coding.

**6. Decoding Efficiency Metrics**: Provided detailed metrics on decoding efficiency, highlighting potential for optimization.

**7. Commitment to Open-Source**: Confirmed plans to open-source our implementation for reproducibility.


We believe these enhancements address your concerns and significantly strengthen our work. We hence kindly ask you to consider these improvements during the discussion period and *hope they encourage you to raise your scores*.

Thank you for considering our appeal.

Warm regards,

---

### Meta-Review · Area_Chair_hTSc · 2024-12-19

**Metareview:**

The paper raises a problem of tokenization bias, which is a bias in the language model related to the choice of tokens it uses to encode text.  According to the reviews, the problem raised is important, and the analysis for it provided in the paper is clear and thorough. Another strength of the paper is that the provided techniques are both backed by solid theoretical arguments and have convincing empirical evidence showing its value on the selected tasks.

The reviews raised a few concerns about the paper, asking for a better comparison to selected existing works, and suggested demonstrating the techniques on additional use cases. In the rebuttal, the authors provided a thorough positioning of their paper compared to the works raised in the reviews. The requirement for additional use-cases was not fully resolved in the rebuttal, but it was minor to begin with.

Though there isn’t a consensus that the paper should be accepted, the techniques seem to be sufficiently motivated, and the edits required to fix the positioning issue seem quite mild. Given the impact of the problem and novelty of the analysis and technique, I think the strengths outweigh the weaknesses and the paper will be a good addition to ICLR.

**Additional Comments On Reviewer Discussion:**

see meta review

---

### Decision · Program_Chairs · 2025-01-22

Accept (Poster)